# Distinct neurexin-cerebellin complexes control AMPA- and NMDA-receptor responses in a circuit-dependent manner

Jinye Dai[1,2]*[†], Kif Liakath-Ali[2], Samantha Rose Golf[2], Thomas C Südhof[1,2]*

[1]Howard Hughes Medical Institute, Stanford University, Stanford, United States; [2]Department of Molecular and Cellular Physiology, Stanford University, Stanford, United States

*For correspondence:
jinye.dai@mssm.edu (JD);
tcs1@stanford.edu (TCS)

Present address: [†]Department of Pharmacological Sciences and Neuroscience, Icahn School of Medicine at Mount Sinai, New York, United States

Competing interest: The authors declare that no competing interests exist.

**Abstract** At CA1→subiculum synapses, alternatively spliced neurexin-1 (Nrxn1$^{SS4+}$) and neurexin-3 (Nrxn3$^{SS4+}$) enhance NMDA-receptors and suppress AMPA-receptors, respectively, without affecting synapse formation. Nrxn1$^{SS4+}$ and Nrxn3$^{SS4+}$ act by binding to secreted cerebellin-2 (Cbln2) that in turn activates postsynaptic GluD1 receptors. Whether neurexin-Cbln2-GluD1 signaling has additional functions besides regulating NMDA- and AMPA-receptors, and whether such signaling performs similar roles at other synapses, however, remains unknown. Here, we demonstrate using constitutive Cbln2 deletions in mice that at CA1→subiculum synapses, Cbln2 performs no additional developmental roles besides regulating AMPA- and NMDA-receptors. Moreover, low-level expression of functionally redundant Cbln1 did not compensate for a possible synapse-formation function of Cbln2 at CA1→subiculum synapses. In exploring the generality of these findings, we examined the prefrontal cortex where Cbln2 was recently implicated in spinogenesis, and the cerebellum where Cbln1 is known to regulate parallel-fiber synapses. In the prefrontal cortex, Nrxn1$^{SS4+}$-Cbln2 signaling selectively controlled NMDA-receptors without affecting spine or synapse numbers, whereas Nrxn3$^{SS4+}$-Cbln2 signaling had no apparent role. In the cerebellum, conversely, Nrxn3$^{SS4+}$-Cbln1 signaling regulated AMPA-receptors, whereas now Nrxn1$^{SS4+}$-Cbln1 signaling had no manifest effect. Thus, Nrxn1$^{SS4+}$- and Nrxn3$^{SS4+}$-Cbln1/2 signaling complexes differentially control NMDA- and AMPA-receptors in different synapses in diverse neural circuits without regulating synapse or spine formation.

## Editor's evaluation

This manuscript examines signaling complexes involving neurexins and cerebellins that bridge the two sides of the synaptic junction. Through carefully executed experiments, the study shows that the basic framework of the complexes operates broadly across different synapses in the brain albeit with subtle differences. This work is of broad interest to neuroscientists studying mechanisms regulating synapse formation and maintenance.

## Introduction

Synaptic organizers are cell-adhesion molecules that direct the formation of synapses and the specification of synapse properties (*Siddiqui and Craig, 2011*; *Ribic and Biederer, 2019*; *Sanes and Zipursky, 2020*; *Südhof, 2021*; *Kim et al., 2021*; *Graham and Duan, 2021*). Multiple candidate synaptic organizers were described, among which neurexins and their multifarious ligands are arguably the best studied (reviewed in *Noborn and Sterky, 2021*; *Gomez et al., 2021*; *Südhof, 2017*; *Kasem et al., 2018*). Neurexins are presynaptic adhesion molecules encoded in mice by the *Nrxn1*,

*Nrxn2*, and *Nrxn3* genes. Each neurexin gene directs synthesis of longer α-neurexins and shorter β-neurexins from separate promoters (*Tabuchi and Südhof, 2002*). In addition, the *Nrxn1* gene (but not the *Nrxn2* and *Nrxn3* genes) contains a third promoter for the even shorter Nrxn1γ (*Sterky et al., 2017*). Neurexin transcripts are extensively alternatively spliced at multiple sites, resulting in thousands of neurexin isoforms whose expression is tightly regulated (*Lukacsovich et al., 2019*; *Nguyen et al., 2016*; *Ullrich et al., 1995*; *Fuccillo et al., 2015*). Among the sites of alternative splicing of neurexins, splice site 4 (SS4) is possibly the most important because it regulates the interaction of neurexins with many ligands, including cerebellins (reviewed in *Südhof, 2017*).

Cerebellins are secreted multimeric C1q-domain proteins that in mice are encoded by four genes (*Cbln1-4*), and that function as trans-synaptic adaptors by connecting presynaptic neurexins to postsynaptic receptors (reviewed in *Yuzaki, 2018*; *Matsuda, 2017*). Cbln1, Cbln2, and Cbln4 are broadly expressed in brain where they are synthesized in restricted distinct subsets of neurons, whereas Cbln3 is specific for cerebellar granule cells and requires Cbln1 for secretion (*Bao et al., 2006*; *Miura et al., 2006*; *Seigneur and Südhof, 2017*). For example, cerebellar granule cells express high levels of Cbln1 but only modest levels of Cbln2, excitatory entorhinal cortex (EC) neurons express predominantly Cbln4, and medial habenula (mHb) neurons express either Cbln2 or Cbln4 (*Seigneur and Südhof, 2017*; *Liakath-Ali et al., 2022*). Although all cerebellins bind to presynaptic neurexins, they interact with different postsynaptic receptors: Cbln1 and Cbln2 bind to GluD1 and GluD2 (*Matsuda et al., 2010*), whereas Cbln4 binds to neogenin-1 (Neo1) and DCC (*Wei et al., 2012*; *Haddick et al., 2014*; *Zhong et al., 2017*). Deletion of Cbln4 in the EC or of Neo1 in the dentate gyrus (DG) blocks long-term potentiation at EC→DG synapses, but does not change the number or basal synaptic transmission at these synapses (*Liakath-Ali et al., 2022*). By connecting presynaptic neurexins to postsynaptic GluDs (Cbln1 and Cbln2) or to Neo1/DCC (Cbln4), cerebellins are thought to mediate trans-synaptic signaling and to organize synapses, but their precise functions are incompletely understood.

In the cerebellum (which is where cerebellins were first studied – hence the name!), deletion of Cbln1 or of its receptor GluD2 (gene symbol *Grid2*) throughout development causes a partial loss of parallel-fiber synapses and a complete loss of long-term plasticity (*Hirai et al., 2005*; *Uemura et al., 2007*; *Rong et al., 2012*). In GluD2 KO mice, parallel-fiber synapses develop initially at least in part, but subsequently decline, with a 40–50% decrease in adult GluD2 KO mice (*Kashiwabuchi et al., 1995*; *Kurihara et al., 1997*; *Takeuchi et al., 2005*). These observations gave rise to the notion that cerebellins may be involved in synapse formation, even though only a fraction of synapses are lost, whereas synaptic plasticity is completely ablated.

Analyses of genetic deletions of *Cbln1*, *Cbln2*, and *Cbln4* outside of the cerebellum revealed behavioral changes and abnormal synaptic transmission, but generally caused little or no synapse loss (*Kusnoor et al., 2010*; *Rong et al., 2012*; *Otsuka et al., 2016*; *Seigneur et al., 2018*; *Seigneur et al., 2018*; *Seigneur et al., 2021*), consistent with a role for cerebellins in shaping synapse properties. For example, constitutive Cbln1/2 double and Cbln1/2/4 triple KO mice displayed major behavioral impairments but no synapse loss at 2 months of age, although synapse numbers declined in the striatum and prefrontal cortex (PFC) over the next 4–6 months (*Seigneur and Südhof, 2018*). Furthermore, constitutive deletion of Cbln2 produced obsessive-compulsive behaviors in mice that resulted from insufficient activation of serotonergic neurons in the dorsal raphe and could be reversed by administration of serotonergic agonists (*Seigneur et al., 2021*). Similarly, conditional deletion of Cbln2 in the mHb led to major behavioral alterations and a rapid decline in mHb→interpeduncular nucleus synaptic transmission, but produced synapse loss only after 3 months (*Seigneur et al., 2018*).

Viewed together, these studies suggested that in multiple brain regions, cerebellins are essential for regulating synaptic properties, are not involved in the initial formation of synapses, but are required for long-term stability of synapses. However, several studies challenged these conclusions. Specifically, RNAi-induced suppression of GluD1 expression was found to suppress excitatory synapse formation in the hippocampal CA1 region, as revealed by analyses of AMPAR/NMDAR EPSCs and of the spine density in biolistic-transfected hippocampal slice cultures, or in acute slices of the adolescent hippocampus (*Tao et al., 2018*). These effects required the GluD1 ligand Cbln2, which is puzzling since Cbln2 expression appears to be absent from CA1 or CA3 region neurons and the constitutive Cbln2 KO did not produce an apparent synapse loss in the CA1 region at 1–2 months of age (*Seigneur et al., 2018*; *Seigneur and Südhof, 2017*). In addition, deletion of neurexins or of GluD1 in the hippocampal formation also did not decrease synapse numbers (*Dai et al., 2021*;

*Missler et al., 2003*; *Aoto et al., 2015*). In another study, an RNAi-induced knockdown of *Cbln4* in cortical pyramidal neurons of the somato-sensory cortex caused an inhibitory synapse loss via a GluD1-dependent mechanism (*Fossati et al., 2019*), which is also puzzling since Cbln4 is known to bind to Neogenin-1 and DCC, while biophysical studies showed that it does not bind to GluD2 (*Zhong et al., 2017*), which is homologous to and has the same function as GluD1 (*Dai et al., 2021*). In a third study, slightly elevated expression of human Cbln2 in mouse PFC was reported to increase the spine density, implying an increase in synapse formation (*Shibata et al., 2021*). However, this observation also raised questions because in *Cbln2* KO mice of the same age, little synapse loss is detected in the cortex (*Seigneur et al., 2018*). Even in the cerebellum of *Cbln1* KO mice, the observed synapse loss is not accompanied by an equivalent decrease in spine density (*Hirai et al., 2005*), and it is unknown whether 'naked' spines form by itself or represent the remnants of synapses that have been lost. Moreover, the increase in spine density in the PFC in the (*Shibata et al., 2021*) paper was larger than the increase in Cbln2 expression. Finally, a synthetic synaptic organizer protein composed of Cbln1 fused to neuronal pentraxin 1 was shown to induce synapse formation in vivo (*Suzuki et al., 2020*), but in this experiment the binding partners of the synthetic protein were unclear, especially since little is known about the function of neuronal pentraxin 1, and the nature of the synaptogenic activity remained unexplored. Thus, presently available data raise multiple questions that need to be addressed for further progress.

We previously demonstrated that at CA1→subiculum synapses, presynaptic neurexin-1 containing an insert in SS4 (Nrxn1$^{SS4+}$) dominantly enhanced NMDA-receptor (NMDAR) EPSCs, whereas presynaptic neurexin-3 containing an insert in SS4 (Nrxn3$^{SS4+}$) potently suppressed AMPA-receptor (AMPAR) EPSCs (*Aoto et al., 2013*; *Dai et al., 2019*). More recently, we showed that Nrxn1$^{SS4+}$ and Nrxn3$^{SS4+}$ both act by binding to Cbln2, which in turn binds to GluD1 and GluD2 (*Dai et al., 2021*). Thus at CA1→subiculum synapses, Nrxn1/3$^{SS4+}$-Cbln2-GluD1 complexes mediate trans-synaptic signaling that controls NMDARs and AMPARs (*Dai et al., 2021*). No changes in synapse density were detected as a function of any of these manipulations – in fact, the massive increase in AMPAR EPSCs induced by the Cbln2 deletion suggested that if a change in synapses had occurred, it should have been an increase, not a decrease, in synapse numbers (*Dai et al., 2021*).

These results characterized a trans-synaptic signaling pathway that organized a specific synaptic circuit (CA1→subiculum synapses). However, only CA1→subiculum synapses were studied, and only after they had fully developed, raising multiple questions. Does *Cbln2* have essential roles at CA1→subiculum synapses in addition to its AMPAR/NMDAR-regulatory function in adult brain? Since low levels of Cbln1 are also present at these synapses, is it possible that Cbln1 compensates for such additional functions in mature synapses? Furthermore, does Cbln2 perform identical functions at different subtypes of CA1→subiculum synapses, where the properties of synapses formed on regular- and burst-firing neurons are quite different (*Wójtowicz et al., 2010*; *Wozny et al., 2008a*; *Wozny et al., 2008b* )? More broadly and possibly more importantly, does a signaling pathway similar to the Nrxn1/3$^{SS4+}$-Cbln2-GluD1 pathway at CA1→subiculum synapses operate at other synapses in brain, or is this pathway specific to CA1→subiculum synapses? To address these questions, we here first compared the contributions of Cbln1 and Cbln2 in different CA1→subiculum synapses and probed their function in relation to upstream Nrxn1$^{SS4+}$ and Nrxn3$^{SS4+}$ signals. We then studied the potential role of Nrxn1/3$^{SS4+}$-Cbln1/2 signaling in two other paradigmatic synapses, namely Layer 2/3→Layer 5/6 excitatory synapses in the mPFC and parallel-fiber synapses in the cerebellum, which we investigated because previous work demonstrated a role for cerebellins in these brain regions. Our data suggest that (1) the Nrxn1/3$^{SS4+}$-Cbln1/2 signaling pathway has no role in synapse or spine formation but functions to shape the NMDAR- and AMPAR-content at multiple types of synapses in diverse circuits, and (2) that different types of synapses exhibit distinct facets of this overall signaling pathway, such that in the mPFC, only the Nrxn1$^{SS4+}$-Cbln2 signaling mechanism is present, whereas in the cerebellum, only the Nrxn3$^{SS4+}$-Cbln1 signaling pathway operates.

## Results

### Constitutive deletion of Cbln2 suppresses NMDARs and enhances AMPARs both at regular- and at burst-firing subiculum neuron synapses

Previous conclusions that presynaptic Nrxn1[SS4+] and Nrxn3[SS4+] regulate postsynaptic NMDARs and AMPARs, respectively, via binding to Cbln2, but that Nrxn1[SS4+], Nrxn3[SS4+], and Cbln2 are not required for synapse formation relied on conditional manipulations at mature CA1→subiculum synapses (*Dai et al., 2021*). In contrast to these results, studies in the cerebellum (*Hirai et al., 2005*; *Ito-Ishida et al., 2008*; *Rong et al., 2012*; *Yuzaki, 2011*) and the PFC (*Shibata et al., 2021*) suggested a function for Cbln1 and Cbln2, respectively, in synapse formation, raising the question whether conditional deletions might have overlooked a developmental synapse formation role of Cbln2 at CA1→subiculum synapses. Moreover, previous experiments did not differentiate between CA1→subiculum synapses on regular- and on burst-firing neurons that exhibit distinct forms of long-term plasticity (*Wozny et al., 2008b*). To explore whether Cbln2 may have an earlier developmental role in addition to its regulation of AMPARs and NMDARs at mature CA1→subiculum synapses, we examined the effect of a constitutive deletion of Cbln2. To determine whether Cbln2 may have distinct functions at synapses on regular- and burst-firing neurons, moreover, we studied these synapses separately at the same time (*Figure 1*).

We generated littermate WT and constitutive Cbln2 KO mice and examined CA1→subiculum synaptic transmission in acute slices at postnatal day 35–42 (P35-42) (*Figure 1A*). In these experiments, we distinguished between regular- or burst-firing neurons in the subiculum by their electrical properties, stimulated axons emanating from the CA1 region, the major source of excitatory inputs into the subiculum (*Böhm et al., 2018*), and monitored EPSCs. In both regular- and burst-firing neurons, the constitutive Cbln2 deletion caused a large elevation (~50%) in AMPAR-EPSC amplitudes and a similarly large decrease (~50%) in NMDAR-EPSC amplitudes, as quantified in input/output curves to control for differences in stimulation efficiency (*Figure 1B, C, G and H*). Moreover, the coefficient of variance of evoked AMPAR-EPSCs did not change, suggesting that the constitutive Cbln2 deletion did not greatly alter the release probability (*Figure 1—figure supplement 1*). These results duplicate those obtained with conditional deletions (*Dai et al., 2021*), suggesting that the absence of Cbln2 throughout development did not produce an additional change in synapses. Furthermore, the finding that synapses on regular- and burst-firing neurons, the two different major types of excitatory synapses in the subiculum, are identically regulated by Cbln2 was confirmed in additional conditional deletion experiments (*Figure 1—figure supplement 2*).

CA1→subiculum synapses on regular- and burst-firing subiculum neurons exhibit distinct forms of LTP, with the former expressing an NMDAR-dependent form of postsynaptic LTP, whereas the latter displays a presynaptic form of LTP (*Wozny et al., 2008b*). The Cbln2 deletion had no effect on presynaptic LTP in burst-firing neuron synapses (*Figure 1D and E*), which undergo a characteristic change in paired-pulse ratios (PPRs) after LTP induction (*Figure 1F*). However, the Cbln2 deletion abolished postsynaptic LTP in regular-firing neurons (*Figure 1I and J*) without a change of PPR after induction (*Figure 1K*). Thus, the Cbln2 deletion produces the same change in AMPAR- and NMDAR-EPSCs in burst- and regular-firing subiculum neurons, but selectively ablates NMDAR-dependent postsynaptic LTP in regular-firing subiculum neurons without affecting presynaptic LTP in burst-firing neurons. Consistent with the dramatic changes in AMPAR- and NMDAR-responses in *Cbln2* KO mice, we observed significant impairments in contextual learning and memory in *Cbln2* KO mice as monitored using the two-chamber avoidance test (*Figure 2*; see also *Dai et al., 2019*).

The finding that the constitutive and conditional deletions of Cbln2 produce the same synaptic phenotype suggests that the constitutive deletion, like the conditional deletion, does not impair synapse formation, as would also be indicated by the dramatic increase in AMPAR-EPSC amplitudes induced by the Cbln2 deletion in both conditions. However, since cerebellins are broadly thought to mediate synapse formation (*Kusnoor et al., 2010*; *Mishina et al., 2012*; *Matsuda, 2017*; *Seigneur and Südhof, 2018*; *Yuzaki, 2018*), we examined the overall synapse density in the subiculum as a function of the constitutive Cbln2 deletion using measurements of immunocytochemical staining intensity for vGluT1 and quantifications of synaptic protein levels as a proxy (*Figure 3*). The constitutive Cbln2 KO caused no change in vGluT1 staining intensity (*Figure 3B and C*) or in the levels of multiple synaptic proteins as assessed by quantitative immunoblotting (*Figure 3D and E*). Together with the lack of a decrease in AMPAR-mediated responses, these findings suggest that the constitutive

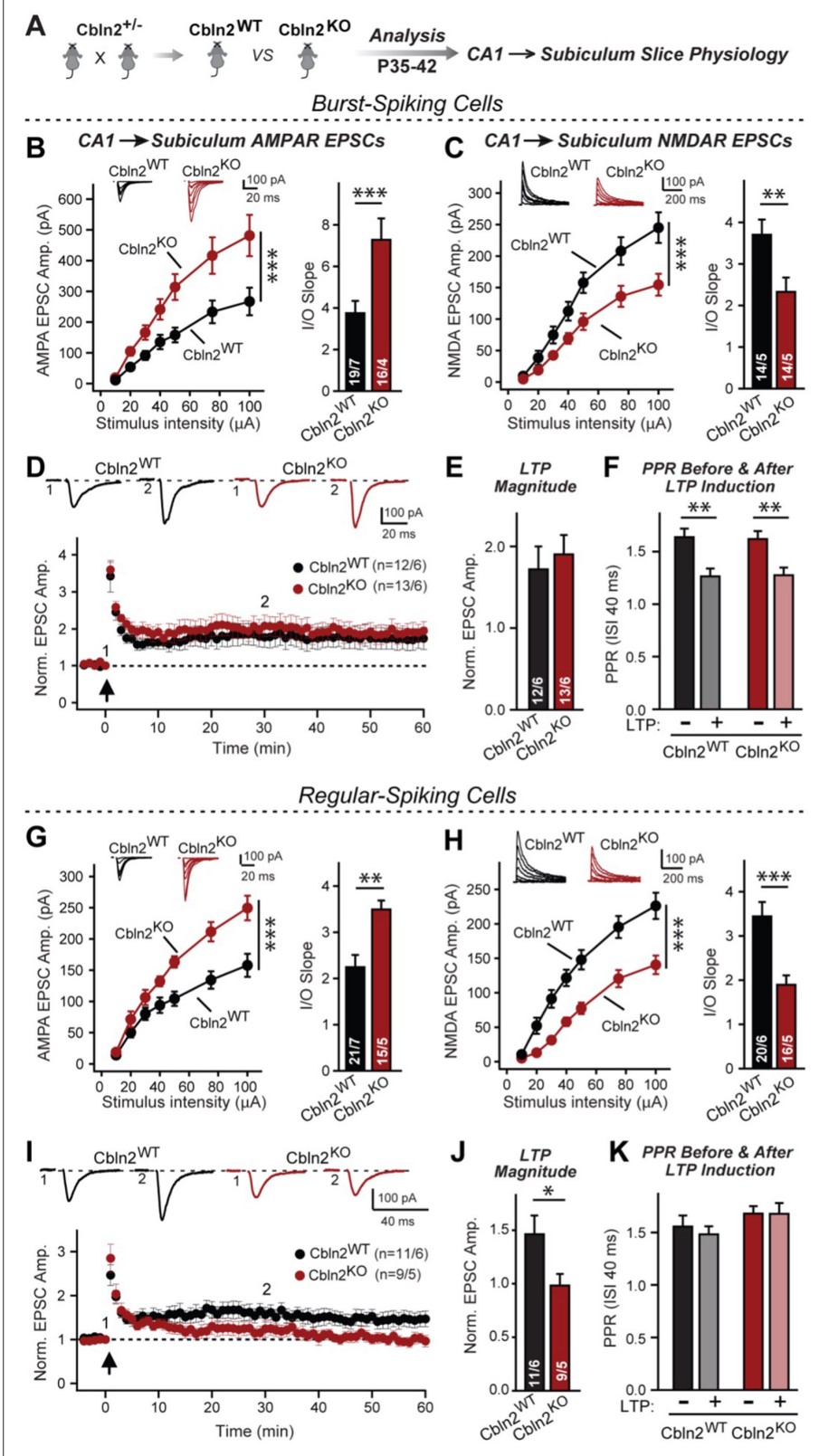

**Figure 1.** Constitutive *Cbln2* deletion increases AMPAR-EPSCs and suppresses NMDAR-EPSCs at CA1→subiculum synapses formed on both burst- and regular-spiking subiculum neurons, and additionally blocks NMDAR-dependent LTP in regular-spiking neurons without affecting cAMP-dependent LTP in burst-spiking neurons. (**A**) Experimental strategy for analysis of littermate wild-type and constitutive *Cbln2* KO mice. (**B**) & (**C**) Input/output

*Figure 1 continued on next page*

*Figure 1 continued*

measurements of evoked AMPAR- and NMDAR-EPSCs recorded from burst-spiking neurons in acute subiculum slices reveal that the *Cbln2* KO enhances AMPAR-EPSCs (**B**) but suppresses NMDAR-EPSCs (**C**). EPSCs were evoked by stimulation of CA1 axons in acute slices from littermate control and *Cbln2* KO mice at P35-42 (left, summary plots of input-output curves with sample traces on top; right, summary graph of input/output slopes). (**D–F**) The *Cbln2* KO had no effect on the presynaptic LTP typical for burst-spiking neurons that is induced by four 100 Hz/1 s stimulus trains with 10 s intervals under voltage-clamp (D, summary plot of AMPAR-EPSC amplitudes with sample traces on top; E, summary graph of the LTP magnitude (normalized EPSC amplitudes during the last 5 min of recordings at least 30 min after LTP induction); F, summary graph of paired-pulse ratios before and after LTP induction as a measure of the release probability). (**G**) (**H**) Same as **B & C**, but recorded from regular-spiking neurons. Note that the AMPAR-EPSC and NMDAR-EPSC phenotype of the *Cbln2* KO is identical in burst- and regular-spiking neurons. (**I–K**) The *Cbln2* KO abolishes NMDAR-dependent postsynaptic LTP that is typical for regular-firing subiculum neurons, and does not involve a change in PPR. Data are from experiments analogous to those described in D-F. All data are means ± SEM. Number of neurons/mice are indicated in bars. Statistical significance was assessed by unpaired two-tailed t-test or two-way ANOVA (*$p \leq 0.05$, **$p \leq 0.01$, and ***$p \leq 0.001$).

The online version of this article includes the following source data and figure supplement(s) for figure 1:

**Source data 1.** Constitutive Cbln2 deletion increases AMPAR-EPSCs and suppresses NMDAR-EPSCs at CA1→subiculum synapses formed on both burst- and regular-spiking subiculum neurons, and additionally blocks NMDAR-dependent LTP in regular-spiking neurons without affecting cAMP-dependent LTP in burst-spiking neurons.

**Figure supplement 1.** Constitutive *Cbln2* deletion does not alter the coefficient of variation of AMPAR-EPSCs in burst- and regular-spiking subiculum neurons (corresponding to *Figure 1*).

**Figure supplement 1—source data 1.** Constitutive Cbln2 deletion does not alter the coefficient of variation of AMPAR-EPSCs in burst- and regular-spiking subiculum neurons.

**Figure supplement 2.** Conditional *Cbln2* KO in the subiculum produces the same phenotype as the constitutive *Cbln2* KO at the two different types of CA1→subiculum synapses that are formed on burst- and regular-spiking neurons.

**Figure supplement 2—source data 1.** Conditional Cbln2 KO in the subiculum produces the same phenotype as the constitutive Cbln2 KO at the two different types of CA1→subiculum synapses that are formed on burst- and regular-spiking neurons.

---

ablation of Cbln2 expression throughout development, similar to the conditional deletion in juvenile mice, does not decrease synapse numbers.

## Cbln2 regulates AMPARs and NMDARs via a trans-synaptic Nrxn1$^{SS4+}$- and Nrxn3$^{SS4+}$-dependent mechanism, respectively

We next set out to test whether the constitutive Cbln2 KO phenotype is due to the ablation of normally occurring presynaptic Nrxn1$^{SS4+}$ and Nrxn3$^{SS4+}$ signals, as suggested by previous studies (*Aoto et al., 2013*; *Dai et al., 2019* and *Dai et al., 2021*). Quantifications of the alternative splicing of neurexins at SS4 in the CA1 region, subiculum, PFC, and cerebellum suggest that in the cerebellum, all neurexins are primarily expressed at SS4 +splice variants, whereas in the other three regions examined neurexins are expressed as a mixture of SS4 +and SS4- splice variants (*Figure 4—figure supplement 1*). Thus, a shift in alternative splicing of neurexins at SS4 could play a major regulatory role, as suggested previously (*Ding et al., 2017*; *Fuccillo et al., 2015*; *Iijima et al., 2011*). Therefore we used two experimental paradigms to induce such a shift and thereby to ask whether deletion of Cbln2 blocked the ability of Nrxn1$^{SS4+}$ to enhance NMDAR-EPSCs and of Nrxn3$^{SS4+}$ to suppress AMPAR-EPSCs that have been shown in previous studies (*Aoto et al., 2013*; *Dai et al., 2019*).

First, we crossed constitutive Cbln2 KO mice with conditional Nrxn1$^{SS4+}$ or Nrxn3$^{SS4+}$ knockin mice (*Aoto et al., 2013*; *Dai et al., 2019*), and bilaterally infected the CA1 region of these double-mutant mice by stereotactic injections with AAVs encoding ΔCre (which retains the SS4 +splice variant) or Cre (which converts the presynaptic SS4 +splice variant into the SS4- variant) (*Figure 4A*). The Cbln2 deletion completely ablated the effect of the presynaptic Nrxn1$^{SS4+}$ or Nrxn3$^{SS4+}$ knockin on NMDAR- and AMPAR-ESPCs, respectively (*Figure 4B and C*). None of these manipulations altered PPRs, documenting that they did not influence the release probability (*Figure 4D and E*). These results confirm

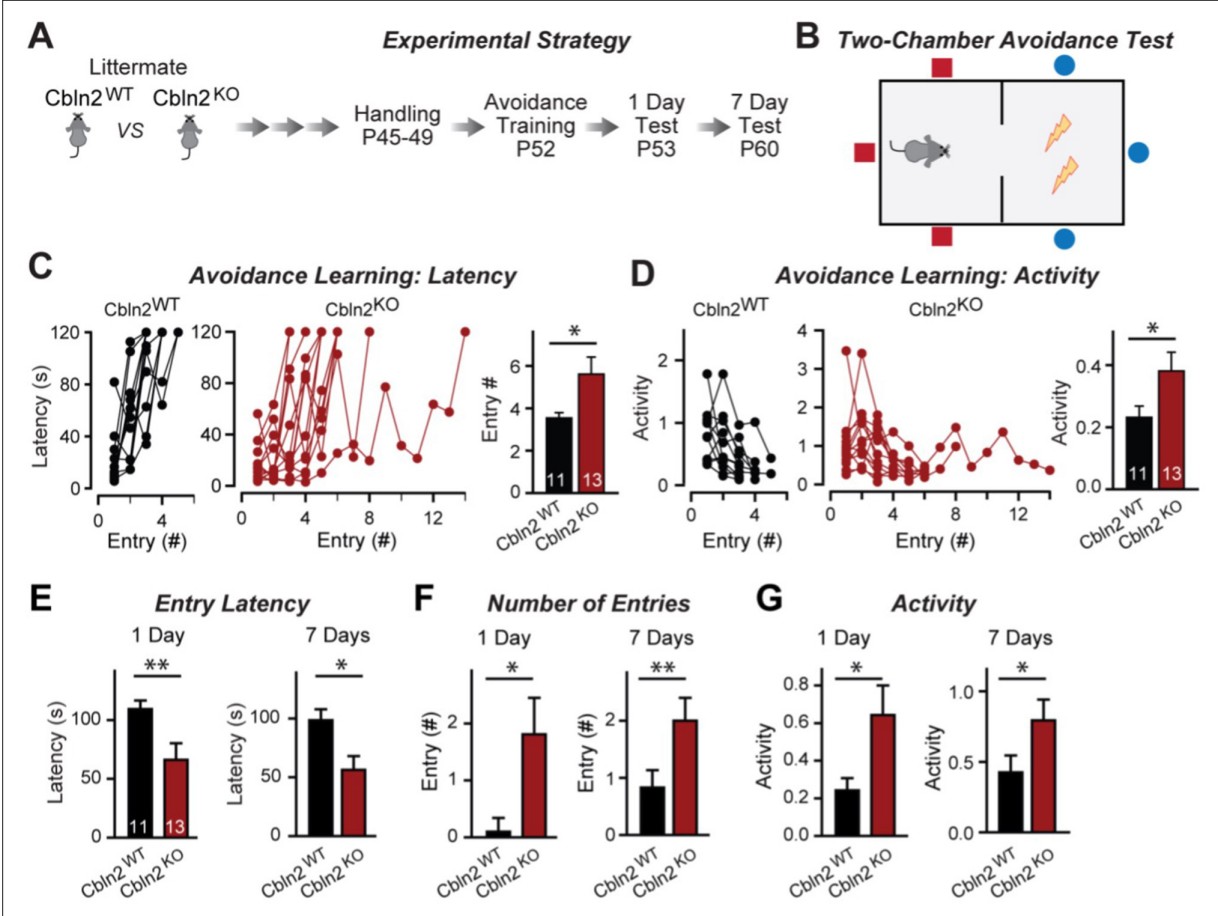

**Figure 2.** Constitutive *Cbln2* deletion impairs contextual memory in the two-chamber avoidance test. (**A**, **B**) Experimental strategy of behavioral experiments utilizing littermate *Cbln2* KO and WT mice (**A**) and design of the two-chamber avoidance test in which mice receive mild electric foot shocks in the otherwise preferred darker chamber (**B**; *Cimadevilla et al., 2001*; *Qiao et al., 2014*). (**C**, **D**) *Cbln2* KO mice exhibit a delayed learning curve during two-chamber avoidance training. Mice learn to stay in the safe space by remembering visual cues to avoid the foot shock (C, trials for each mouse taking to learn when they remain in safe chamber for more than 2 min) (called latency; summary graphs shows number of entries); **D**, graphs for the movement activity measured by four independent infrared photobeams in the safe chamber in 2 min and the summary graph shows activity level in the safe chamber for the last training trial. (**E–G**) *Cbln2* KO severely decreases contextual memory in mice as measured by the two chamber avoidance test 1 day (left graphs) or 7 days (right graphs) after training (summary graphs of E, entry latencies; F, number of entries, and G, mouse activity). Data are means ± SEMs, the number of mice analyzed are depicted in the bars. Statistical analyses were performed by one-tail t-test (\*=p ≤ 0.05; \*\*=p ≤ 0.01).

The online version of this article includes the following source data for figure 2:

**Source data 1.** Constitutive Cbln2 deletion impairs contextual memory in the two-chamber avoidance test.

that Cbln2 is required for transduction of the presynaptic Nrxn1[SS4+] or Nrxn3[SS4+] signals into postsynaptic NMDAR and AMPAR responses, respectively.

Second, we overexpressed Nrxn1β[SS4+] or Nrxn3β[SS4+] in the presynaptic CA1 region in constitutive Cbln2 KO mice in vivo using stereotactic bilateral injections of AAVs (*Figure 4F*). We previously showed that overexpression of Nrxn1β[SS4+] in wild-type CA1 neurons increases NMDAR- but not AMPAR-EPSCs at CA1→subiculum synapses, whereas overexpression of Nrxn3β[SS4+] in wild-type CA1 neurons suppresses AMPAR- but not NMDAR-EPSCs in the same synapses (*Dai et al., 2019*). When we tested the effect of Nrxn1β[SS4+] or Nrxn3β[SS4+] in constitutive Cbln2 KO mice, however, Nrxn1β[SS4+] no longer increased NMDAR-EPSCs and Nrxn3β[SS4+] no longer suppressed AMPAR-EPSCs (*Figure 4G and H*). None of these manipulations altered PPRs, demonstrating that they did not affect presynaptic properties (*Figure 4I and J*). Viewed together, these data suggest that Cbln2 transduces presynaptic Nrxn1[SS4+] and Nrxn3[SS4+] signals into distinct postsynaptic receptor responses at CA1→subiculum synapses.

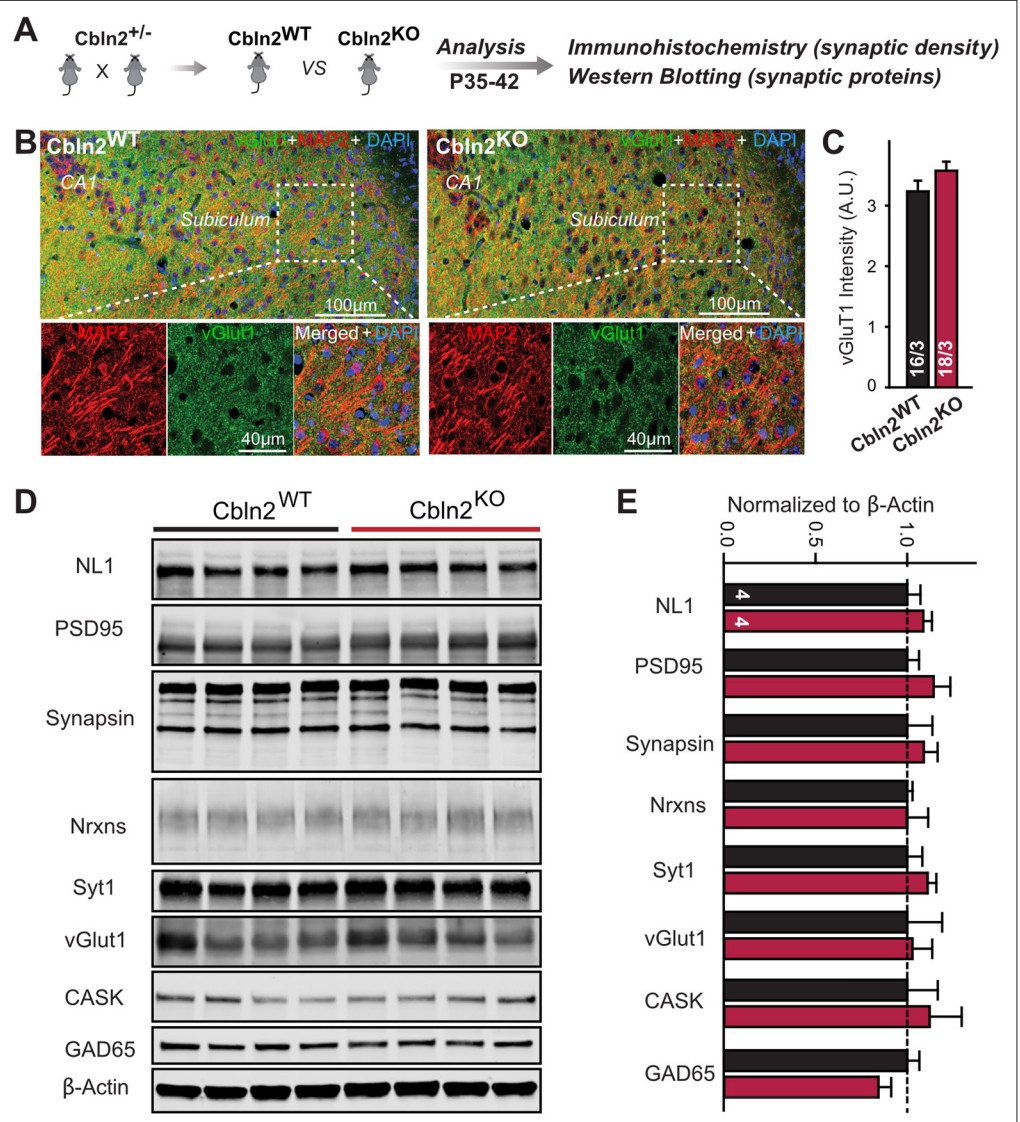

**Figure 3.** Constitutive *Cbln2* deletion does not alter the overall synapse density in the subiculum. (**A**) Experimental strategy for the analysis of littermate wild-type and constitutive *Cbln2* KO mice. (**B**) Representative images of subiculum sections stained for vGluT1 as a proxy of synapse density, MAP2 as a proxy of neuronal density, and DAPI. (**C**) The *Cbln2* KO does not cause a major loss of excitatory synapses in the subiculum as indicated by the vGluT1 staining intensity. (**D & E**) The *Cbln2* KO also does not significantly alter the levels of synaptic proteins in the hippocampus. Protein levels were measured in hippocampal lysates by quantitative immunoblotting using fluorescent secondary antibodies (**D**) representative blots, please also see original full-sized immunoblots in **Figure 3—source data 1**; (**E**), summary graph (levels are normalized for β-actin as an internal standard, and then to the controls to render results from multiple experiments comparable; n=3 independent experiments). Data are means ± SEMs, the number of slices/mice or number of mice analyzed are depicted in the bars; statistical analyses by unpaired two-tailed t-test revealed no significant differences.

The online version of this article includes the following source data for figure 3:

**Source data 1.** Constitutive Cbln2 deletion does not alter the overall synapse density in the subiculum.

## Double deletion of Cbln1 and Cbln2 produces the same phenotype as deletion of Cbln2 alone

Up to this point, our results indicate that Cbln2 functions both at regular- and at burst-firing neuron synapses in the subiculum to control AMPARs and NMDARs without being required for synapse formation. However, in these and earlier experiments we only studied Cbln2, but quantifications show that

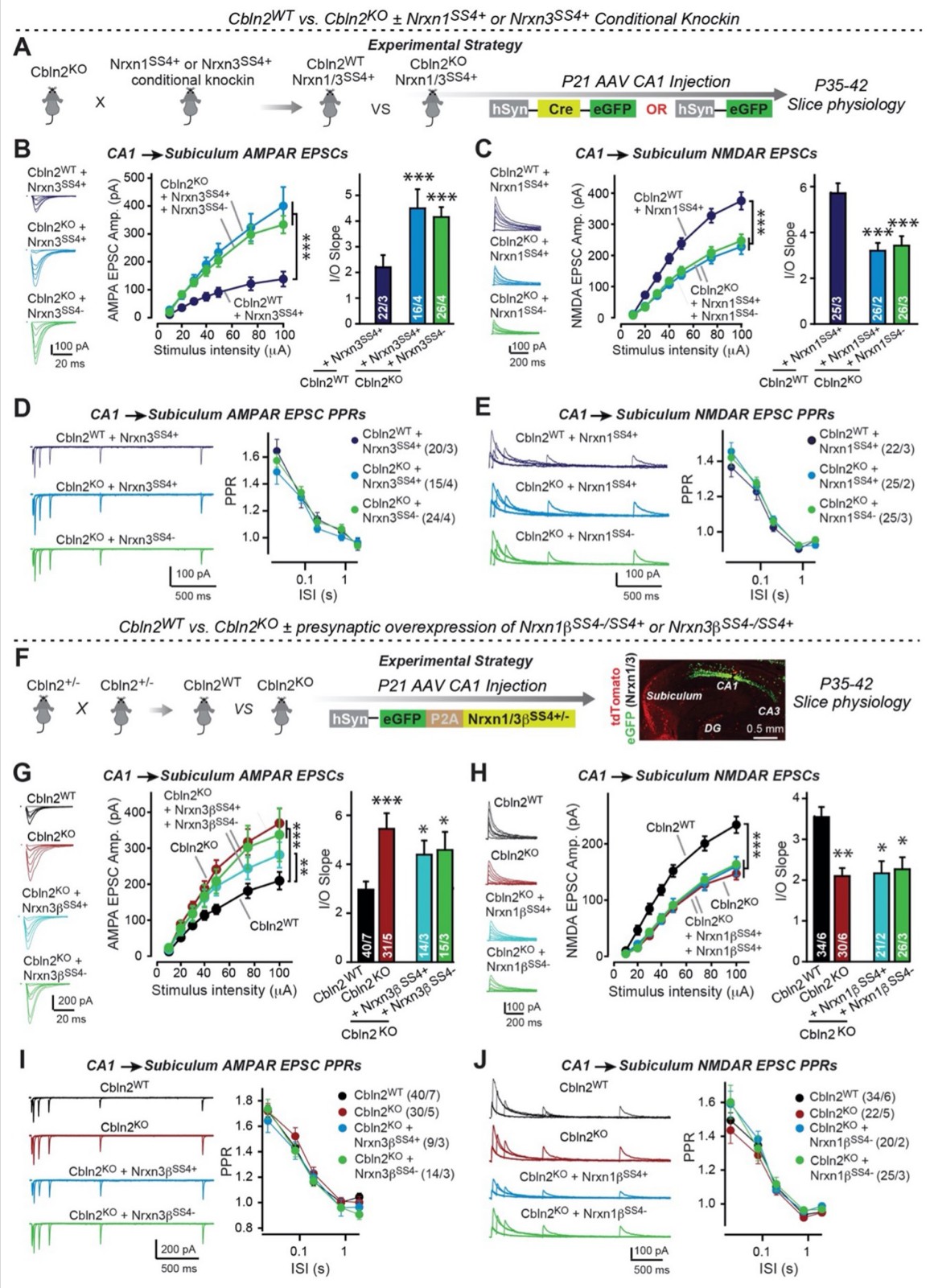

**Figure 4.** Constitutive *Cbln2* deletion occludes regulation of postsynaptic AMPAR- and NMDAR-EPSCs by presynaptic Nrxn1[SS4+] and Nrxn3[SS4+], respectively. (A) Experimental strategy for testing whether the *Cbln2* deletion blocks the effects of Nrxn1[SS4+] and Nrxn3[SS4+] signaling. Constitutive *Cbln2* KO mice (Cbln2[KO]) were crossed with Nrxn1[SS4+] and Nrxn3[SS4+] knockin mice that constitutively express Nrxn1[SS4+] and Nrxn3[SS4+] splice variants, but that are converted into constitutively expressing Nrxn1[SS4-] and Nrxn3[SS4-] splice variants by Cre-recombinase (*Dai et al., 2019*). Three groups of

*Figure 4 continued on next page*

*Figure 4 continued*

mice were compared: 1. Cbln2$^{WT}$ mice expressing Nrxn1$^{SS4+}$ or Nrxn3$^{SS4+}$, 2. Cbln2$^{KO}$ mice expressing Nrxn1$^{SS4+}$ and Nrxn3$^{SS4+}$ in which presynaptic CA1 neurons were infected stereotactically at P21 with AAVs expressing inactive ΔCre (retains presynaptic Nrxn1$^{SS4+}$ and Nrxn3$^{SS4+}$ genotype); and 3. Nrxn1$^{SS4+}$ and Nrxn3$^{SS4+}$ in which presynaptic CA1 neurons were infected stereotactically at P21 with AAVs expressing active Cre (generates presynaptic Nrxn1$^{SS4-}$ and Nrxn3$^{SS4-}$ genotype). CA1→subiculum synapses were then analyzed in acute slices from these mice at P35-42. (B) On the background of the *Cbln2* KO, knockin of Nrxn3$^{SS4+}$ no longer suppresses AMPAR-ESPCs, nor does it reverse the increase in AMPAR-EPSCs induced by the Cbln2 KO at CA1→subiculum synapses (left, representative traces; middle, summary plot of the input/output relation; right, summary graph of the slope of the input/output relations). (C) Similarly, Nrxn1$^{SS4+}$ no longer enhances NMDAR-ESPCs on the background of the *Cbln2* KO, nor does it reverse the decrease in NMDAR-EPSCs induced by the Cbln2 KO (left, representative traces; middle, summary plot of the input/output relation; right, summary graph of the slope of the input/output relations). (D & E) Constitutive expression of Nrxn1$^{SS4+}$ and Nrxn3$^{SS4+}$ alone or in combination with the *Cbln2* KO have no effect on the paired-pulse ratio of evoked AMPAR-EPSCs (D) or NMDAR-EPSCs (E) at CA1→subiculum synapses (left, sample traces; right, summary plots of PPRs). (F) Alternative experimental strategy for testing whether the *Cbln2* deletion blocks the effects of Nrxn1$^{SS4+}$ and Nrxn3$^{SS4+}$ signaling. Analysing the epistatic relation of neurexin alternative splicing at SS4 with the *Cbln2* KO at CA1→subiculum synapses using viral overexpression of Nrxn1β$^{SS4+}$ or Nrxn3β$^{SS4+}$ in *Cbln2* KO mice. The CA1 region of constitutive *Cbln2* KO mice was bilaterally infected at P21 by stereotactic injections with AAVs expressing Nrxn1β$^{SS4+}$, Nrxn1β$^{SS4-}$, Nrxn3β$^{SS4+}$, or Nrxn3β$^{SS4-}$, and subiculum neurons were analyzed 2–3 weeks later. The representative image on the right depicts the signal for eGFP (which is co-expressed with the neurexins) in CA1 neurons after 2 weeks infection. (G) On the background of the *Cbln2* KO, overexpression of Nrxn3β$^{SS4+}$ again no longer suppresses AMPAR-ESPCs, nor does it reverse the increase in AMPAR-EPSCs induced by the *Cbln2* KO at CA1→subiculum synapses (left, representative traces; middle, summary plot of the input/output relation; right, summary graph of the slope of the input/output relations). (H) Similarly, overexpressed Nrxn1β$^{SS4+}$ no longer enhances NMDAR-ESPCs on the background of the *Cbln2* KO, nor does it reverse the decrease in NMDAR-EPSCs induced by the Cbln2 KO (left, representative traces; middle, summary plot of the input/output relation; right, summary graph of the slope of the input/output relations). (I & J ) Overexpression of any neurexin has no effect on the paired-pulse ratio of evoked AMPAR-EPSCs (I) or NMDAR-EPSCs (J) (left, sample traces; right, summary plots of PPRs). Data are means ± SEM. Number of neurons/mice are indicated in bars. Statistical significance was assessed by unpaired two-tailed t-test comparing to control and two-way ANOVA (*$p \leq 0.05$, **$p \leq 0.01$, and ***$p \leq 0.001$).

The online version of this article includes the following source data and figure supplement(s) for figure 4:

**Source data 1.** Constitutive Cbln2 deletion occludes regulation of postsynaptic AMPAR- and NMDAR-EPSCs by presynaptic Nrxn1$^{SS4+}$ and Nrxn3$^{SS4+}$, respectively.

**Figure supplement 1.** Analysis of neurexin SS4 alternative splicing reveals that whereas all neurexins are almost exclusively expressed as SS4 +variants in the cerebellum, a mixture of SS4 +and SS4- variants is observed in other brain regions.

**Figure supplement 1—source data 1.** Analysis of neurexin SS4 alternative splicing reveals that whereas all neurexins are almost exclusively expressed as SS4 +variants in the cerebellum, a mixture of SS4 +and SS4- variants is observed in other brain regions.

**Figure supplement 1—source data 2.** Analysis of neurexin SS4 alternative splicing reveals that whereas all neurexins are almost exclusively expressed as SS4 +variants in the cerebellum, a mixture of SS4 +and SS4- variants is observed in other brain regions.

Cbln1 is also expressed in the subiculum, albeit at much lower levels (*Figure 5—figure supplements 1 and 2*). Moreover, the constitutive Cbln2 KO does not alter the expression of Cbln1, Nrxns, and GluDs (*Figure 5—figure supplement 2*). Cbln1 and Cbln2 have nearly indistinguishable biochemical properties, suggesting that they are functionally redundant. The finding that Cbln1 is also expressed in the subiculum raises the possibility that the observed Cbln2 KO phenotype reflects only those Cbln2 functions that are most sensitive to a decrease in overall Cbln1/2 levels, and that the remaining Cbln1 could occlude other phenotypes. To address this possibility, we generated conditional Cbln1/2 double KO mice and analyzed the effect of the double Cbln1/2 deletion in the subiculum by electrophysiology, using an expansive array of measurements to ensure that no effects were overlooked (*Figure 5A*).

Measurements of NMDAR-EPSCs and AMPAR-ESPCs at CA1→subiculum synapses revealed the same phenotype in Cbln1/2 double conditional KO as the conditional and constitutive Cbln2-only deletion, namely a decrease in NMDAR-responses and an increase in AMPAR-responses (*Figure 5B and D*). These phenotypes were validated using input/output measurements to control for variabilities in the position of the stimulating electrode, and were due to a postsynaptic mechanism, as described before, since PPRs did not change (*Figure 5C and E*). We also measured spontaneous mEPSCs as an indirect measure of synaptic activity and synapse numbers, and monitored mEPSCs at two holding potentials (–70 mV and +60 mV) to capture the contributions of both AMPARs and NMDARs to mEPSCs (*Figure 5F–I*). mEPSCs monitored at –70 mV are exclusively mediated by AMPARs, whereas mEPSCs monitored at +60 mV contain contributions of both AMPAR and NMDAR activation. At both holding potentials, the mEPSC frequency was massively enhanced (~100–130% increase) by the Cbln1/2 double KO, presumably because of the increased AMPAR-responses leads to increased

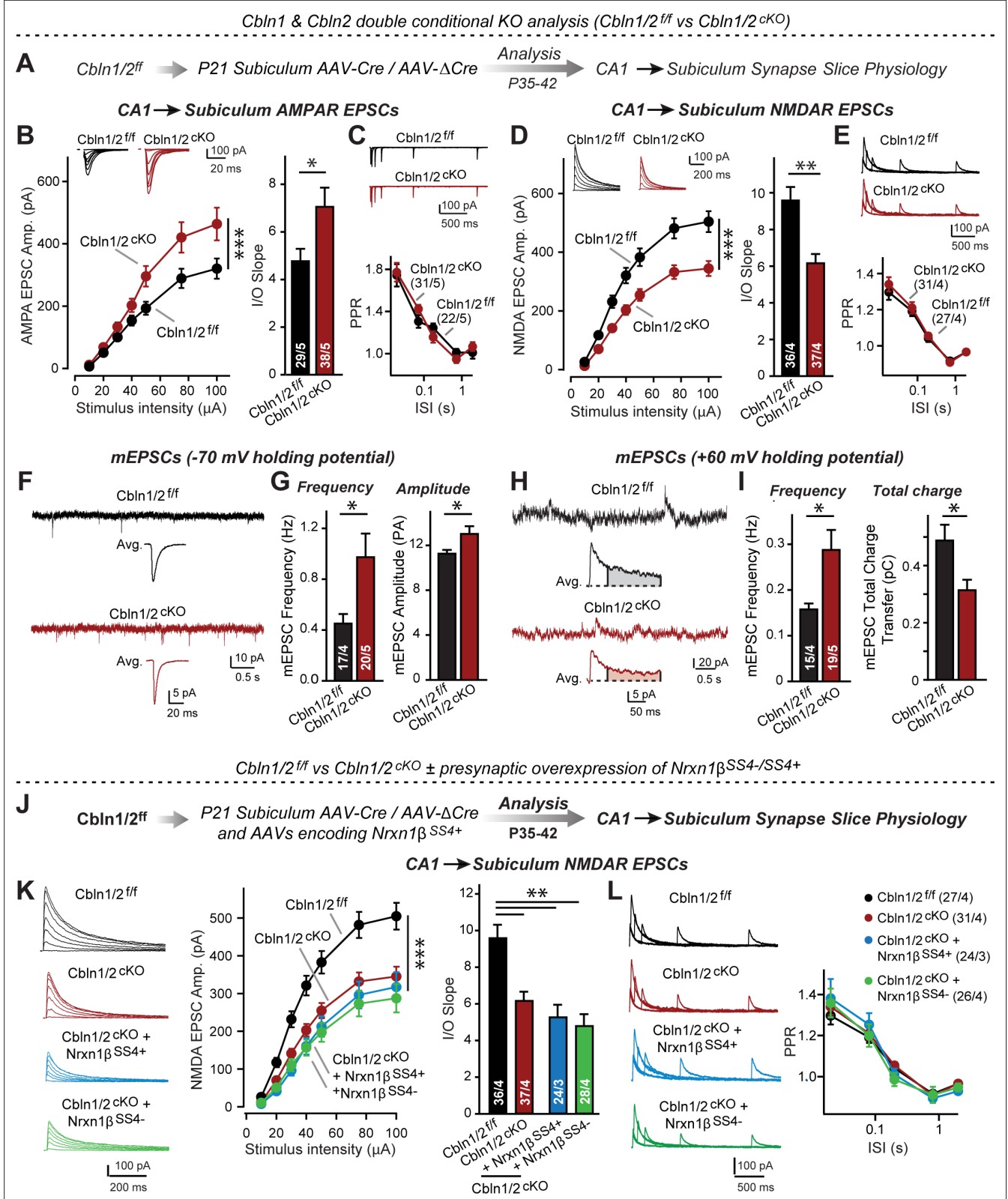

**Figure 5.** *Cbln1* and *Cbln2* double KO in the subiculum phenocopies the *Cbln2* single KO at CA1→subiculum synapses. (**A**) Experimental strategy. AAVs encoding Cre or ΔCre (as a control) were stereotactically injected into the subiculum of conditional KO mice at P21, and mice were analyzed by slice physiology 2–3 weeks later. (**B-E**) Input/output measurements of evoked EPSCs recorded from combined burst- and regular-spiking neurons in acute subiculum slices reveal that the conditional Cbln2 KO enhances AMPAR-EPSCs (**B**) without changing the paired-pulse ratio of AMPAR-EPSCs (**C**) but

*Figure 5 continued on next page*

*Figure 5 continued*

suppresses NMDAR-EPSCs (**D**), again without changing the paired-pulse ratio of NMDAR-EPSCs (**E**). Sample traces are shown above the respective summary plots and graphs. (**F-I**) Analyses of mEPSCs recorded at –70 mV and +60 mV holding potentials from burst- and regular-firing neurons in the subiculum after deletion of both Cbln1 and Cbln2 reveal an increase in mEPSC frequency measured at both holding potentials, but a decrease in charge transfer only of mEPSCs monitored at a+60 mV holding potential consistent with the decreased NMDAR-EPSC amplitude detected during input/output measurements (**F**, sample traces; **G**, bar graphs of the mEPSC frequency and amplitude, respectively; **H & I**, same as **F & G** but for recordings at +60 mV). (**J**) Experimental strategy. The subiculum region of Cbln1/2$^{cKO}$ was bilaterally infected at P21 by stereotactic injections of AAVs expressing ΔCre-eGFP (Cbln1/2$^{f/f}$) or Cre-eGFP (Cbln1/2$^{cKO}$), and then two weeks later cohorts of mice injected with Cre were further injected into the CA1 region with AAVs expressing Nrxn1β$^{SS4+}$ or Nrxn1β$^{SS4-}$. Mice were then analyzed at P49-P56 by acute slice electrophysiology. (**K**) Overexpressed Nrxn1β$^{SS4+}$ no longer enhances NMDAR-ESPCs on the background of the double Cbln1/2 cKO, nor does it reverse the decrease in NMDAR-EPSCs induced by the double Cbln1/2 cKO (left, representative traces; middle, summary plot of the input/output relation; right, summary graph of the slope of the input/output relations). (**L**) Conditional deletion of both Cbln1 and Cbln2 without or with presynaptic overexpression of Nrxn1β$^{SS4+}$ or Nrxn1β$^{SS4-}$ does not alter paired-pulse ratios of NMDAR EPSCs. Left panels show sample traces; right panels summary plots of the paired-pulse ratio as a function of the interstimulus interval. Data are means ± SEMs; the number of cells/mice are depicted in the bars. Statistical analyses were performed by two-way ANOVA or unpaired two-tailed t-test comparing KOs to WT (*p≤0.05; **p≤0.01, and ***p≤0.001).

The online version of this article includes the following source data and figure supplement(s) for figure 5:

**Source data 1.** Cbln1 and Cbln2 double KO in the subiculum phenocopies the Cbln2 single KO at CA1→subiculum synapses.

**Figure supplement 1.** Analyses of the region-specific expression patterns of *Cbln1* and *Cbln2* using single-molecule RNA in situ hybridization (A & B).

**Figure supplement 2.** The constitutive *Cbln2* deletion does not alter the overall expression levels of Nrxn1, Nrxn2, Nrxn3, Cbln1, Cbln2, GluD1 and GluD2 mRNAs in the PFC or subiculum.

**Figure supplement 2—source data 1.** The constitutive Cbln2 deletion does not alter the overall expression levels of Nrxn1, Nrxn2, Nrxn3, Cbln1, Cbln2, GluD1 and GluD2 mRNAs in the PFC or subiculum.

detection of mEPSCs at both holding potentials. Importantly, the average mEPSC amplitude was increased at the –70 mV holding potential but the average mEPSC total charge transfer decreased at the +60 mV, consistent with the observation that the double Cbln1/2 KO increases AMPAR- but decreases NMDAR-responses (*Figure 5B and D*).

Finally, we asked whether the phenotype of the double Cbln1/2 KO might be more sensitive to manipulations of neurexins than that of the Cbln2 single KO. Focusing on Nrxn1 and NMDARs, we found that as with the single deletion of Cbln2, NMDAR EPSCs were no longer altered upon presynaptic overexpression of Nrxn1β containing or lacking an insert in SS4 (*Figure 5J–L*). Overall, these data suggest that the Cbln1/2 double deletion has the same overall phenotype as the Cbln2 single deletion, with a dramatic change in AMPAR- and NMDAR-EPSCs but no apparent changes in presynaptic release probability.

## Nrxn1$^{SS4+}$-Cbln2 complexes upregulate NMDARs in PFC, but Nrxn3$^{SS4+}$-Cbln2 complexes have no effect

Our studies in two different CA1→subiculum synapses, described here and previously (*Aoto et al., 2013* and 2015; *Dai et al., 2019* and *Dai et al., 2021*), show that Nrxn1$^{SS4+}$-Cbln2 complexes upregulate NMDARs whereas Nrxn3$^{SS4+}$-Cbln2 complexes downregulate AMPARs. Does this trans-synaptic signaling pathway also operate in non-subiculum synapses, or is this a specific feature of subiculum synapses?

To address this question, we conditionally deleted Cbln2 from the mPFC (*Figure 6A*). The mPFC exhibits robust expression of Cbln2 (*Figure 5—figure supplement 1*), and the upregulation of Cbln2 expression in the mPFC in humans was postulated to increase spine numbers and thereby synapse formation specifically in humans (*Shibata et al., 2021*). We stereotactically injected AAVs encoding ΔCre (as a control) or Cre into the mPFC of Cbln2 conditional KO mice at P21, and analyzed layer2/3 (L2/3) → layer5/6 (L5/6) synapses in acute slices at P35-42 (*Figure 6A*). For this purpose, we placed the stimulating electrode close to L2/3 neurons and recorded from L5/6 pyramidal neurons (*Figure 6B*). Strikingly, the conditional Cbln2 KO produced a massive increase (~100%) in the AMPAR/NMDAR ratio of L2/3→L5/6 synaptic transmission in the mPFC (*Figure 6C*). This increase was due to a large reduction (~50%) in NMDAR-EPSC amplitudes, whereas the AMPAR-EPSC amplitudes were not altered (*Figure 6C*). Again, we observed no changes in PPRs (*Figure 6D*).

These data suggest that at L2/3→L5/6 synapses of the adult mPFC, Cbln2 operates only as a regulator of NMDARs but not of AMPARs (*Figure 6C and D*). Is the function of Cbln2 in the mPFC

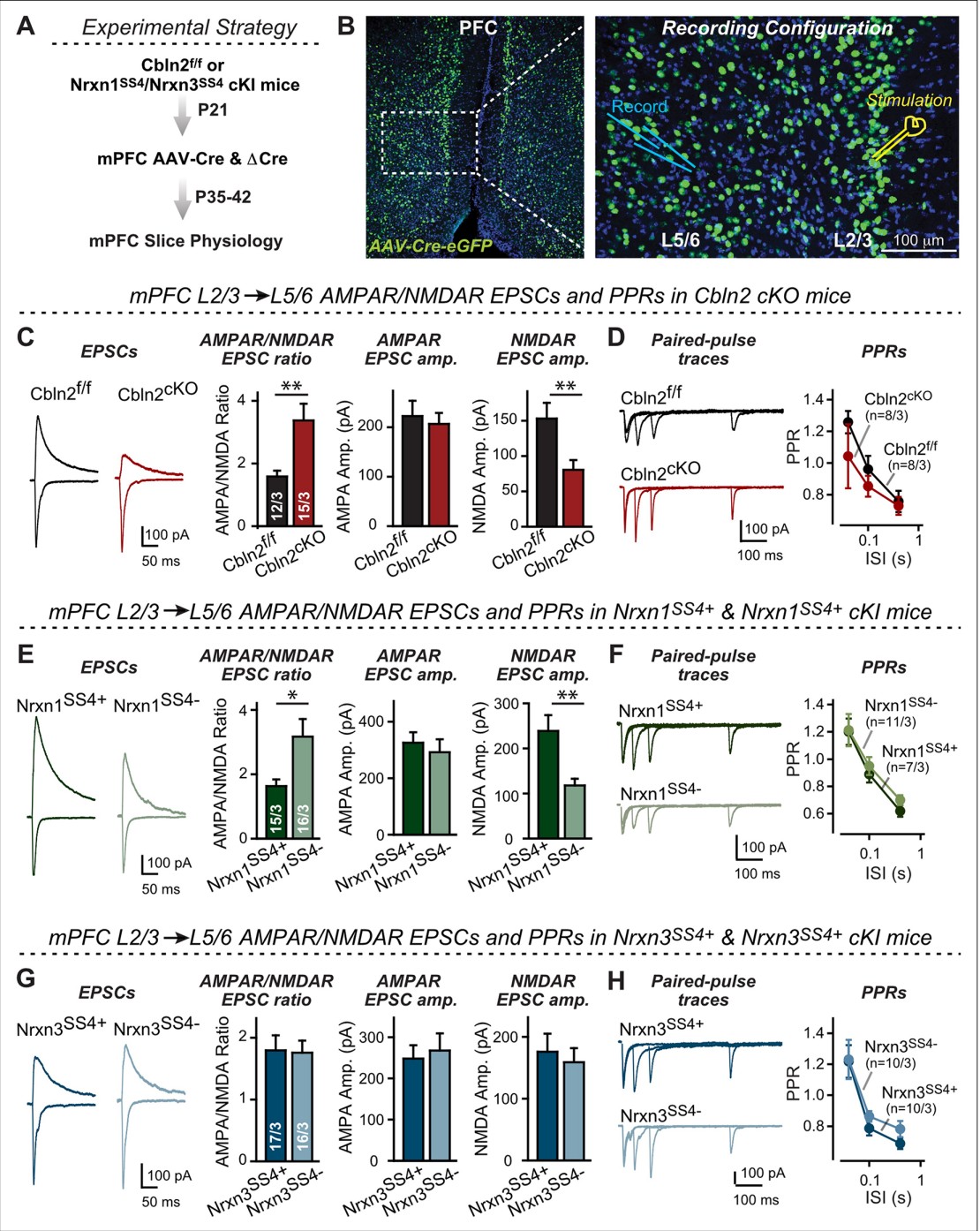

**Figure 6.** Nrxn1SS4+-Cbln2 signaling controls NMDAR-EPSCs but not AMPAR-EPSCs in the PFC, whereas Nrxn3SS4+-Cbln2 signaling does not regulate either AMPAR- or NMDAR-EPSCs in the PFC. (**A & B**) Experimental strategy (left, flow diagram of the experiments); middle and right, Analysis strategies of Cbln2/Nrxn1-SS4/Nrxn3-SS4 conditional KO. Right, the mPFC region of Cbln2cKO was bilaterally infected at P21 by stereotactic injections of AAVs expressing ΔCre-eGFP (Cbln2f/f) or Cre-eGFP (Cbln2cKO), and L5/6 pyramidal neurons in the prelimbic cortex (PL) region were analyzed 2–3 weeks later (**A**) The stimulation electrode was placed in L2/3 as indicated and applied same stimulation intensity/duration for all groups of mice to keep the consistency (**B**) **C**. Left, sample traces of evoked AMPAR- and NMDAR-EPSCs at Cbln2f/f and Cbln2cKO mPFC brain slices; Right, statistics of AMPA/NMDA ratios, AMPAR-EPSCs amplitude, and NMDAR-EPSCs amplitude. (**D**) Left, sample traces of paired-pulse measurements from each condition; Right, summary plots of PPRs. (**E & F**) Same as (**C & D**), but recorded from Nrxn1SS4+ knockin mice in which ΔCre retains a constitutive expression of Nrxn1-SS4+splice variants, whereas Cre converts the Nrxn1-SS4+variants into constitutive Nrxn1-SS4- variants. (**G & H**) Same as (**C & D**) but recorded from Nrxn3SS4+ knockin mice in which ΔCre retains a constitutive expression of Nrxn3-SS4+splice variants, whereas Cre converts the Nrxn3-SS4+variants

*Figure 6 continued on next page*

*Figure 6 continued*

into constitutive Nrxn3-SS4- variants. Data are means ± SEM. Number of neurons/mice are indicated in bars. Statistical significance was assessed by unpaired two-tailed t-test or two-way ANOVA (*$P \leq 0.05$, **$P \leq 0.01$, and ***$p \leq 0.001$).

The online version of this article includes the following source data for figure 6:

**Source data 1.** Nrxn1$^{SS4+}$-Cbln2 signaling controls NMDAR-EPSCs but not AMPAR-EPSCs in the PFC, whereas Nrxn3$^{SS4+}$-Cbln2 signaling does not regulate either AMPAR- or NMDAR-EPSCs in the PFC.

also downstream of neurexins? To examine this question, we investigated the effect of the constitutive expression of Nrxn1$^{SS4+}$ or Nrxn3$^{SS4+}$ at L2/3→L5/6 synapses in the mPFC. We bilaterally infected the mPFC of Nrxn1$^{SS4+}$ or Nrxn3$^{SS4+}$ conditional knockin mice (*Aoto et al., 2013*; *Dai et al., 2019*) by stereotactic injections with AAVs encoding ΔCre (which retains the SS4 +variant) or Cre (which converts SS4 +variants into SS4- variants). Consistent with the Cbln2 KO results, only the constitutive presynaptic expression of Nrxn1$^{SS4+}$ produced a phenotype, whereas the constitutive expression of Nrxn3$^{SS4+}$ had no effect (*Figure 6E–H*). Specifically, constitutive expression of Nrxn1$^{SS4+}$ deletion caused a large increase (~100%) in the AMPAR/NMDAR ratio due to a large decrease (~100%) in the NMDAR-EPSC amplitudes but not AMPAR-EPSC; this phenotype was abolished by conversion of Nrxn1$^{SS4+}$ to Nrxn1$^{SS4-}$ (*Figure 6E*). In contrast, the constitutive expression of Nrxn3$^{SS4+}$ had no effect on the AMPAR/NMDAR ratio or either AMPAR-EPSC or NMDAR-EPSC amplitudes (*Figure 6G*). Again, none of these manipulations altered PPRs, documenting that they did not influence the release probability (*Figure 6F and H*). These results are consistent with the Cbln2 KO findings in the mPFC, validating the Nrxn1$^{SS4+}$→Cbln2→NMDAR signaling pathway in the mPFC in the absence of the Nrxn3$^{SS4+}$→Cbln2→AMPAR signaling pathway that we also observed in the subiculum.

The synaptic changes in conditionally deleted Cbln2 mPFC neurons were likely not due to a decrease in synapse numbers because a decrease in synapse numbers should equally affect AMPAR- and NMDAR-EPSCs. However, a recent prominent study suggested that humans may have a higher spine density (and by proxy, a higher synapse density) in the mPFC than mice because of an increase in Cbln2 expression (*Shibata et al., 2021*). If correct, this result would imply that Cbln2 regulates synapse formation during development and in the adult.

To test this hypothesis and to further confirm whether Cbln2 is involved in synapse formation and/or maintenance in the mPFC, we quantified synapse and spine numbers in constitutive Cbln2 KO mice, using littermate WT controls. We used constitutive instead of conditional Cbln2 KO because Cbln2 expression is absent throughout development in these mice, which mimics the conditions used by *Shibata et al., 2021* in which Cbln2 expression was slightly increased throughout development. Measurements of either the spine density (*Figure 7A-C*) or the synapse density by monitoring both presynaptic (vGlut1) and postsynaptic markers (Homer1) (*Figure 7D and E*) failed to uncover any change in Cbln2 KO mice. The results indicate that in mice, Cbln2 has no role in spinogenesis or synapse formation in the mPFC, but does not exclude the possibility that by an unknown mechanism a modest increase in Cbln2 expression might still significantly increase spine numbers as observed by *Shibata et al., 2021*.

## In the cerebellum, Nrxn3$^{SS4+}$-Cbln1 complexes suppress AMPARs, whereas Nrxn1$^{SS4+}$-Cbln1 complexes have no effect

Cerebellins were discovered in the cerebellum, with constitutive deletions of Cbln1 or of its receptor GluD2 causing a marked but partial loss of parallel-fiber synapses (*Hirai et al., 2005*; *Kashiwabuchi et al., 1995*; *Kurihara et al., 1997*; *Takeuchi et al., 2005*). However, it is unclear whether this synapse loss (that starts after synapses are initially formed) reflects a direct function of Cbln1 in synapse formation or represents an indirect effect of an increase in AMPAR-mediated synaptic transmission to which parallel-fiber synapses may be particularly sensitive (note that parallel-fiber synapses do not express functional NMDARs; *Llano et al., 1991*; *Perkel et al., 1990*). In the first case, Cbln1 would perform a function in the cerebellum that differs from that of Cbln2 in the subiculum and mPFC; in the second case, Cbln1 would also regulate AMPARs in parallel-fiber synapses in a function that would be the same as that of Cbln2 in the subiculum and mPFC, and that should become detectable in synapses after they have been formed.

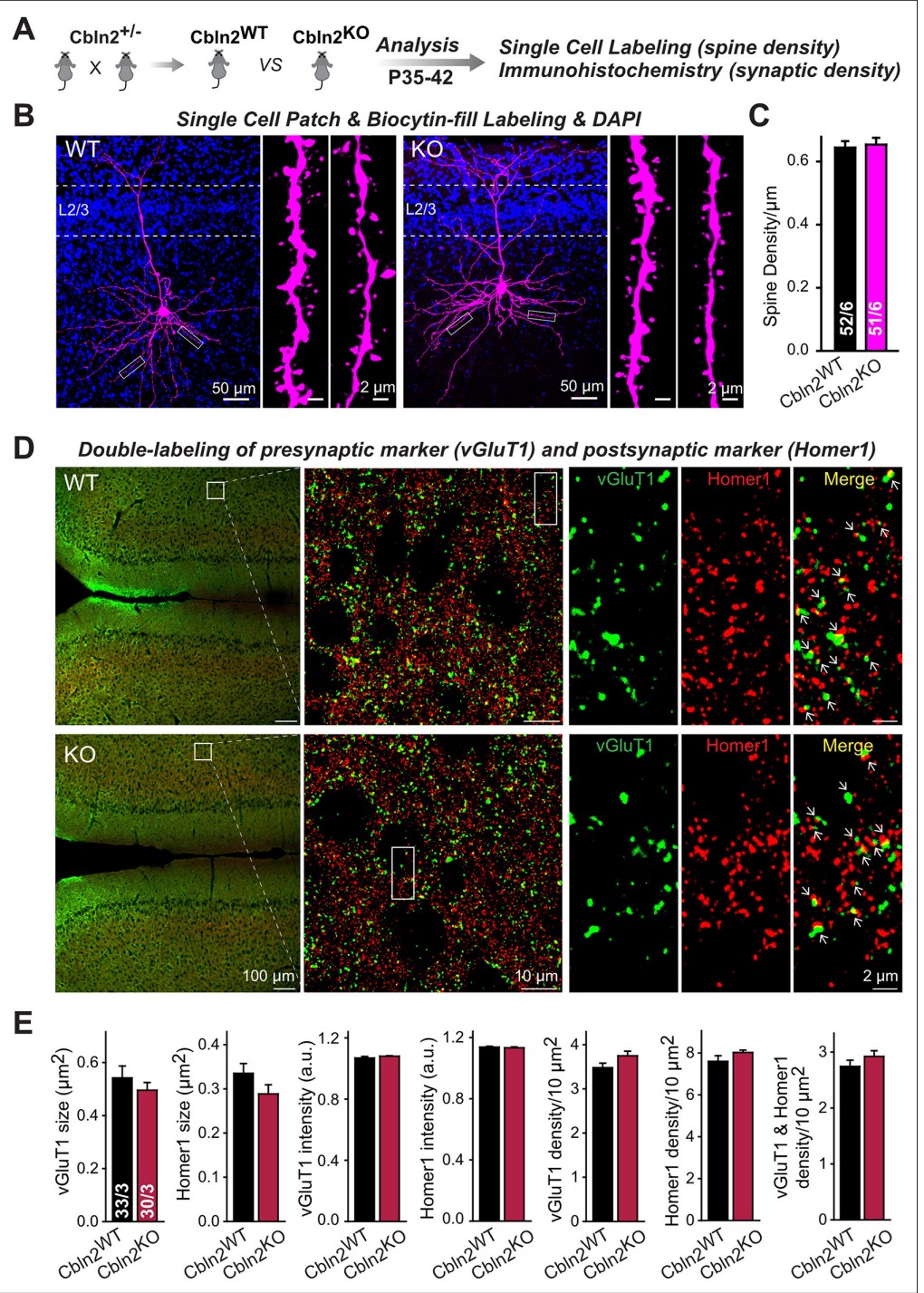

**Figure 7.** Constitutive *Cbln2* deletion does not alter the overall synapse density in the PFC. (**A**) Experimental strategy for the analysis of littermate wild-type and constitutive *Cbln2* KO mice. (**B**) Representative images of PFC sections with a single biocytin-filled neuron (left, 20 x images of PFC sections with biocytin labeling and DAPI staining; right, 100 x images of dendrite spines). (**C**) The *Cbln2* KO does not cause a change of dendrite spine density in the PFC as quantified in biocytin-filled neurons. (**D**) Representative images of PFC sections stained for vGluT1 as a presynaptic marker and Homer1 as a postsynaptic maker (left, 10 x images of PFC sections with vGluT1 and Homer1 staining; middle, 100 x images of the prelimbic cortex (PL), right, enlarged images of vGluT1 staining, Homer1 staining, and their colocalization). (**E**) The *Cbln2* KO also does not significantly alter the size of synaptic puncta, the intensity of synaptic markers, and the density of synapses in the PFC. Data are means ± SEMs, the

*Figure 7 continued on next page*

*Figure 7 continued*

number of dendrites/cells or sections/mice analyzed are depicted in the bars; statistical analyses by unpaired two-tailed t-test revealed no significant differences.

The online version of this article includes the following source data for figure 7:

**Source data 1.** Constitutive Cbln2 deletion does not alter the overall synapse density in the PFC.

To address this question, we stereotactically infected lobes 4–5 of the cerebellum of Cbln1 conditional KO mice at P21 with AAVs encoding ΔCre or Cre, and analyzed synaptic transmission at parallel-fiber synapses at P35-42 (*Figure 8A and B*). Strikingly, the Cbln1 deletion increased the AMPAR-EPSC input/output curve and its slope (*Figure 8C*), without changing the coefficient of variation (CV), indicating that it did not influence the release probability (*Figure 8D*).

These results, based on our analyses of subiculum and mPFC synapses above, imply that Nrxn3$^{SS4+}$-Cbln1 complexes, but not Nrxn1$^{SS4+}$-Cbln1 complexes, control parallel-fiber synapse properties in the cerebellum. Given the fact that both Nrxn1 and Nrxn3 are expressed in the cerebellum almost exclusively as SS4 +splice variants (*Figure 4—figure supplement 1*), this implication is surprising. To validate this conclusion, we again used the mouse lines carrying conditional genetic knockin mutations that cause a constitutive expression of SS4 +variants of endogenous Nrxn1 and Nrxn3. Measurements of parallel-fiber synaptic transmission demonstrated that the presynaptic Nrxn3$^{SS4+}$ knockin fully phenocopied the Cbln1 cKO, whereas the Nrxn1$^{SS4+}$ knockin had no effect (*Figure 8E–H*). As before, none of these manipulations altered the coefficient of variation, suggesting that they did not influence the release probability (*Figure 8F and H*). These results confirm that the function of Cbln1 in cerebellum is dependent on presynaptic Nrxn3$^{SS4+}$ signals and acts to control postsynaptic AMPAR responses at the PF-PC synapses (*Figure 8*).

## Discussion

At CA1→subiculum synapses, signaling by Nrxn1$^{SS4+}$ and Nrxn3$^{SS4+}$ selectively enhances NMDAR-EPSCs and suppresses AMPAR-EPSCs, respectively, via a common mechanism: Binding to Cbln2 that in turn binds to GluD1 (*Dai et al., 2019* and *Dai et al., 2021*). The convergence of distinct Nrxn1$^{SS4+}$ and Nrxn3$^{SS4+}$ signals onto the same Cbln2-GluD1 effectors to produce different downstream effects was unexpected, but was validated by the demonstration that distinct cytoplasmic GluD1 sequences transduce the differential Nrxn1$^{SS4+}$ and Nrxn3$^{SS4+}$ signals (*Dai et al., 2021*). These findings described a trans-synaptic signaling pathway regulating NMDARs and AMPARs, but raised new questions. In particular, given multiple lines of evidence suggesting a role for cerebellins in synapse formation (see Introduction) and given the fact that previous experiments manipulated mature neurons (*Dai et al., 2019* and *Dai et al., 2021*), the question arises whether Cbln2 may have additional functions in synapse formation at CA1→subiculum synapses during development, and whether additional roles of Cbln2 at CA1→subiculum synapses might have been redundantly occluded by low levels of Cbln1 that are present at CA1→subiculum synapses. Even more important, however, is the question whether the Nrxn1$^{SS4+}$- and Nrxn3$^{SS4+}$-Cbln2 signaling pathways (and those of the closely related Cbln1) are specific to CA1→subiculum synapses, or whether they broadly operate in other synapses in brain as well.

We have now addressed these questions. Our data suggest that at CA1→subiculum synapses, Cbln1 does not redundantly occlude a major additional function of Cbln2, that the Nrxn1$^{SS4+}$- and Nrxn3$^{SS4+}$-Cbln2 signaling pathways do not have additional synapse-formation functions during development, and that these signaling pathways are important regulators of NMDARs and AMPARs at multiple types of synapses in the subiculum, PFC, and cerebellum. Strikingly, we show that these signaling pathways do not equally operate at all synapses, but are selectively present in subsets of synapses (*Figure 9*). The evidence supporting these conclusions can be summarized as follows.

First, we showed that a constitutive deletion of Cbln2 operating throughout development has the same effect as the conditional post-developmental deletion of Cbln2 (*Figures 1–3*). Both produced a similar enhancement of AMPAR-EPSCs (up to 100% increase) and suppression of NMDAR-EPSCs (up to 40% decrease), without a change in synapse numbers. Consistent with a broad effect on synapses, the Cbln2 deletion also severely impaired contextual learning (*Figure 2*). Moreover, the constitutive deletion of Cbln2 occluded the dominant effects of Nrxn1$^{SS4+}$ and Nrxn3$^{SS4+}$ signaling on NMDARs and AMPARs, respectively (*Figure 4*), confirming that Nrxn1$^{SS4+}$ and Nrxn3$^{SS4+}$ operate upstream of Cbln2.

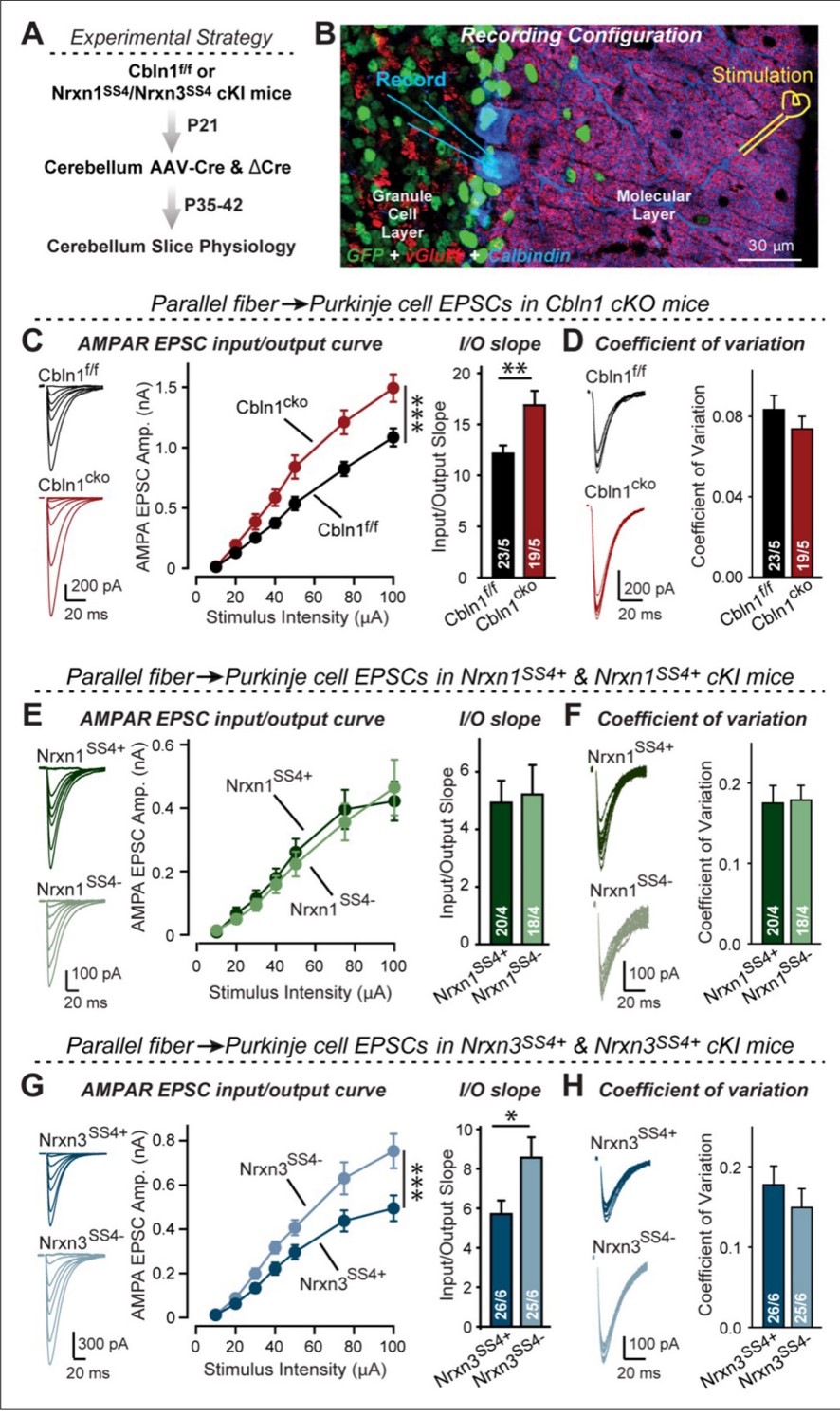

**Figure 8.** Nrxn3^SS4+-Cbln1 signaling controls AMPAR-EPSCs in the cerebellum, but in this brain region Nrxn1^SS4+-Cbln1 signaling has no effect. (**A**) Experimental workflow for analyzing the effect of the *Cbln1* cKO or of the conditional Nrxn1^SS4+ or Nrxn3^SS4+ knockin on parallel-fiber synaptic transmission in the cerebellum. Note that the expression of ΔCre in Nrxn1^SS4+ or Nrxn3^SS4+ knockin mice retains the constitutive expression of their SS4 +splice variants, whereas the expression of Cre converts SS4 +into a constitutive SS4- splice variant. (**B**) Image of a cerebellar cortex section (lobes 4–5) from *Cbln1* cKO mouse in which these lobes were infected at P21 by stereotactic injections of AAVs expressing ΔCre-eGFP (Cbln1^f/f) or Cre-eGFP (Cbln1^cKO). Sections were analyzed at P35 by slice physiology; the positions of the recording electrode in the patched Purkinje cells and of

*Figure 8 continued on next page*

*Figure 8 continued*

the stimulation electrode in the granule cell layer are indicated. (*C*) The *Cbln1* deletion in cerebellum significantly increases the amplitude of AMPAR-EPSCs at parallel-fiber synapses (left, sample traces of evoked AMPAR-EPSCs; middle, summary plot of AMPAR-EPSCs input-output curves; right, summary graph of the slope of AMPAR-EPSC input/output curves). (*D*) The *Cbln1* deletion in cerebellum has no major effect on the coefficient of variation at parallel-fiber synapses, suggesting that it does not greatly change the release probability (left, sample traces of evoked AMPAR-EPSCs with 50 µA stimulus intensity; right, summary graph of the coefficient of variation of AMPAR-EPSCs). (**E & F**) Same as (**C & D**) but recorded from Nrxn1$^{SS4+}$ knockin mice in which ΔCre retains a constitutive expression of Nrxn1-SS4+splice variants, whereas Cre converts the Nrxn1-SS4+variants into constitutive Nrxn1-SS4- variants. (*G & H*) Same as (**E & F**) but for Nrxn3$^{SS4+}$ knockin mice in which ΔCre retains a constitutive expression of Nrxn1-SS4+splice variants, whereas Cre converts the Nrxn1-SS4+variants into constitutive Nrxn1-SS4- variants. Data are means ± SEM. Number of neurons/mice are indicated in bars. Statistical significance was assessed by two-way ANOVA or unpaired two-tailed t-test (*p≤0.05, **p≤0.01, and ***p≤0.001).

The online version of this article includes the following source data for figure 8:

**Source data 1.** Nrxn3$^{SS4+}$-Cbln1 signaling controls AMPAR-EPSCs in the cerebellum, but in this brain region Nrxn1$^{SS4+}$-Cbln1 signaling has no effect.

Second, we examined whether the function of Cbln2 is the same in the two types of CA1→subiculum synapses that are formed on burst- and regular-spiking neurons and that exhibit quite distinct properties (*Wójtowicz et al., 2010*; *Wozny et al., 2008a*; *Wozny et al., 2008b*). In both synapse types, the constitutive and conditional Cbln2 deletion caused the same increase in AMPAR-EPSCs and the same decrease in NMDAR-EPSCs (*Figure 1—figure supplement 1*). The two types of subiculum synapses differ in their forms of LTP (*Wójtowicz et al., 2010*; *Wozny et al., 2008a*; *Wozny et al., 2008b*). Notably, the Cbln2 deletion blocked the NMDAR-dependent LTP of synapses on regular-spiking neurons without affecting the cAMP-dependent LTP in burst-spiking neurons (*Figure 1*). Since the former type of LTP is postsynaptic and latter presynaptic, these findings agree with the conclusion of a postsynaptic regulatory effect of Cbln2 signaling. This deficit in NMDAR-dependent LTP could be due to impaired LTP induction, given the reduced NMDAR-response in Cbln2 KO mice. Alternatively, the deficit in NMDAR-dependent LTP in Cbln2 KO mice could be caused by a a saturation of 'slots' for AMPARs in the postsynaptic specializations, since AMPAR-EPSCs are massively enhanced. However, we previously found that constitutive expression of Nrxn3$^{SS4+}$ suppresses AMPARs without affecting NMDARs, but also blocks NMDAR-dependent LTP (*Aoto et al., 2013*), suggesting that neither an impairment of NMDAR-dependent LTP induction nor a saturation of AMPAR slots is a likely explanation for the loss of LTP in Cbln2-deficient synapses. Alternatively, it is possible that the neurexin→Cbln2-signaling pathway renders postsynaptic specializations competent of responding to NMDAR-dependent LTP induction, although the nature of the signaling mechanism remains unclear.

Third, we investigated the possibility that low-level expression of Cbln1 in the subiculum might redundantly compensate for Cbln2 in additional functions besides regulation of AMPARs and NMDARs, such as synapse formation. Such redundancy may cause an additional function of Cbln2 to become selectively occluded in the Cbln2 KO mice. To explore this possibility, we analyzed Cbln1/2 double KO mice, but identified substantially the same phenotype as in Cbln2 single KO mice (*Figure 5*). Thus, it seems unlikely that low-level expression of Cbln1 prevents manifestation of an additional Cbln2 function.

Fourth, we tested the possible function of the Nrxn1$^{SS4+}$-Cbln2 and Nrxn3$^{SS4+}$-Cbln2 signaling pathways at L2/3→L5/6 synapses in the mPFC, focusing on Cbln2 because it is expressed at higher levels than Cbln1 in the mPFC similar to the subiculum (*Figure 5—figure supplements 1 and 2*). We observed that the Cbln2 deletion caused a suppression of NMDAR-EPSCs, but did not enhance AMPAR-EPSCs (*Figure 6*). This observation suggests that only the Nrxn1$^{SS4+}$-Cbln2 but not the Nrxn3$^{SS4+}$-Cbln2 signaling pathway operates in the mPFC synapses. Consistent with this conclusion, we found that the Nrxn1$^{SS4+}$ switch to Nrxn1$^{SS4-}$ selectively downregulated NMDARs in the PFC because only Nrxn1$^{SS4+}$ but not Nrxn1$^{SS4-}$ can bind to Cbln2 (*Figure 6*). In contrast and different from CA1→subiculum synapses, the Nrxn3$^{SS4+}$ switch to Nrxn3$^{SS4-}$ had no effect on AMPARs in the PFC (*Figure 6*).

Our findings in the PFC are at odds with a recent study reporting that a hominin-specific increase in Cbln2 expression in the PFC induces an elevation of spine numbers, and by implication synapse

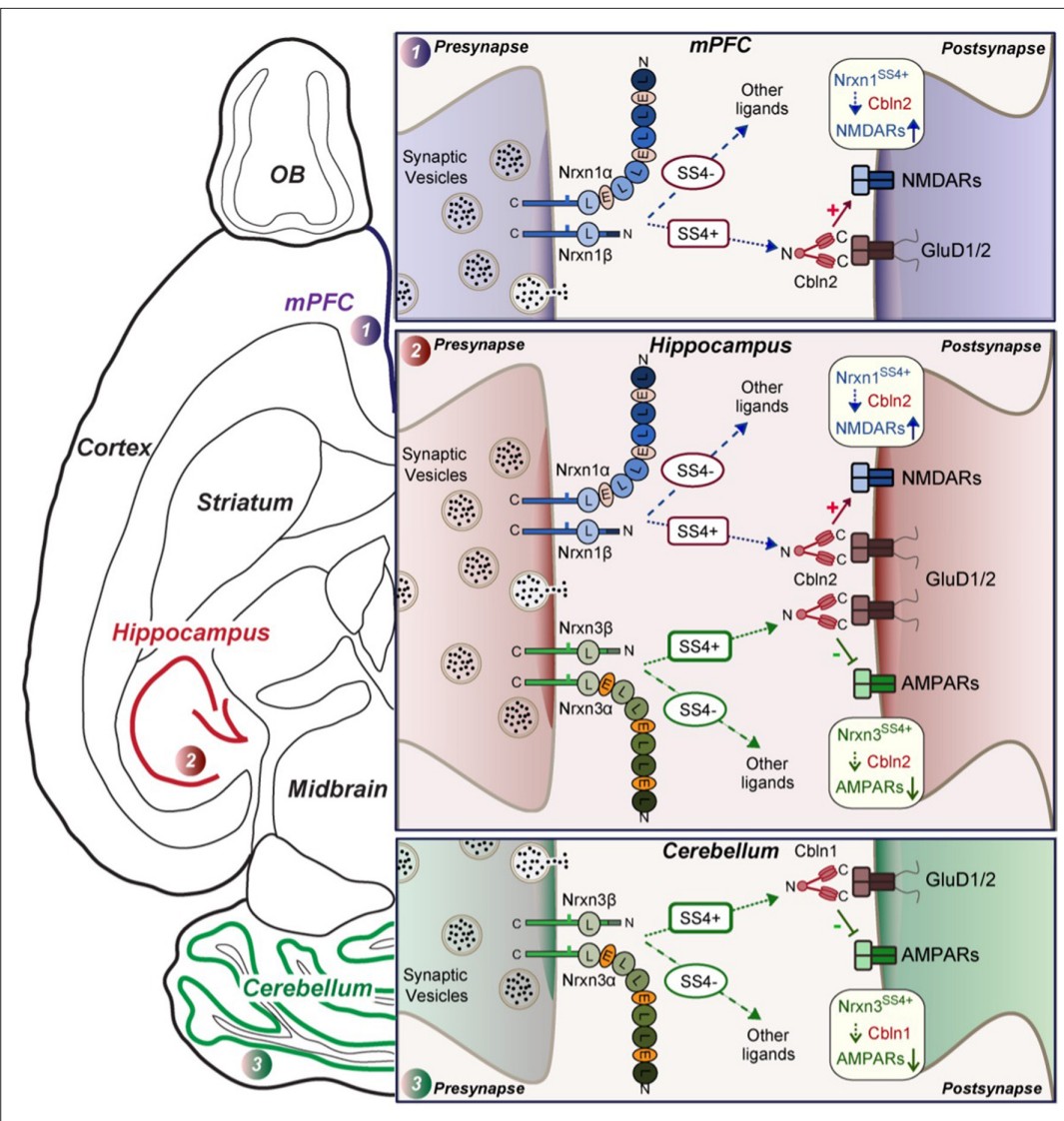

**Figure 9.** Schematic illustration of how Nrxn1$^{SS4+}$-Cbln1/2 and Nrxn3$^{SS4+}$-Cbln1/2 signaling complexes control postsynaptic AMPARs and NMDARs in subicular, prefrontal, and cerebellar circuits. The schematic is based on data shown previously (*Aoto et al., 2013*; *Dai et al., 2019* and *Dai et al., 2021*) and described here. Alternative splicing of presynaptic Nrxn1 and Nrxn3 at SS4 that controls their interactions with Cbln1/2 and thereby with postsynaptic GluD1/2 differentially regulates the postsynaptic content of AMPARs and NMDARs in different brain region. In the hippocampus, Nrxn1$^{SS4+}$-Cbln1/2 complexes enhance NMDAR-EPSCs, whereas Nrxn3$^{SS4+}$-Cbln1/2 complexes suppress AMPAR-EPSCs, with both types of complexes acting via GluD1/2. In the PFC, Nrxn1$^{SS4+}$-Cbln1/2 complexes also enhance NMDAR-EPSCs, but Nrxn3$^{SS4+}$-Cbln1/2 complexes have no effect. In the cerebellum, conversely, Nrxn3$^{SS4+}$-Cbln1/2 complexes suppress AMPAR-EPSCs, whereas now Nrxn1$^{SS4+}$-Cbln1/2 complexes have no effect. These results indicate that Nrxn1$^{SS4+}$-Cbln1/2 and Nrxn3$^{SS4+}$-Cbln1/2 complexes perform universal functions in regulating AMPARs and NMDARs, respectively, but that these regulatory signaling pathways are differentially expressed in different types of synapses.

numbers (*Shibata et al., 2021*). Moreover, *Shibata et al., 2021* showed that a modest increase in Cbln2 expression (~25%) in the mouse PFC leads to a robust increase in spine density (~40%). Although this conclusion is difficult to reconcile with our data, it is possible that overexpression of a protein could have functional effects that are different from those of a loss-of-function. Note, however, that *Shibata et al., 2021* did not explore the effects of a direct genetic manipulation, but instead used a 'humanization' of a putative enhancer for the Cbln2 gene that increases Cbln2 expression. Thus, a plausible alternative explanation for the discrepancy between our results and the conclusions by

*Shibata et al., 2021* is that the enhancer manipulation may have affected other genes besides Cbln2, and that these other genes are responsible for the observed change in spine density.

Fifth and finally, we examined parallel-fiber synapses in the cerebellum, the synapses at which Cbln1 was discovered and which do not express functional NMDARs (*Llano et al., 1991*; *Perkel et al., 1990*). Cbln1 has a well-characterized function at these synapses in maintaining synapse stability and enabling long-term synaptic plasticity (*Kashiwabuchi et al., 1995*; *Kurihara et al., 1997*; *Hirai et al., 2005*; *Takeuchi et al., 2005*). Strikingly, we found that the post-developmental conditional deletion of Cbln1 at these synapses also significantly increased AMPAR-EPSCs similar to the Cbln2 deletion in the subiculum, and that the induced switch from Nrxn3$^{SS4+}$ to Nrxn3$^{SS4-}$ had the same effect on AMPAR-EPSCs as the Cbln1 deletion, whereas the switch from Nrxn1$^{SS4+}$ to Nrxn1$^{SS4-}$ had no effect (*Figure 8*). These experiments suggest that at parallel-fiber synapses of the cerebellum, Nrxn3$^{SS4+}$-Cbln1 signaling controls AMPARs similar to the action of Nrxn3$^{SS4+}$-Cbln2 signaling at CA1→subiculum synapses (*Dai et al., 2019*). At first glance, these results seem to contradict previous studies showing that deletions of Cbln1 or of its GluD2 receptor in cerebellum cause a partial loss of parallel-fiber synapses (reviewed in *Yuzaki, 2018*; *Yuzaki and Aricescu, 2017*). However, this loss affects less than half of all synapses, while all of the remaining synapses are dysfunction and unable to undergo LTD. More importantly, this loss only occurs after an initial formation of at least some synapses (*Kurihara et al., 1997*). Furthermore, the GluD2 deletion was also shown to induce an increase in AMPARs at parallel-fiber synapses (*Yamasaki et al., 2011*), replicating our observations with the Cbln1 deletion since Cbln1 is the major binding partner to GluD2 (*Figure 8*). Viewed together, these results are consistent with the notion that Cbln1 also regulates AMPARs at parallel-fiber synapses, and that the ablation of such regulation causes secondary synapse elimination.

*Figure 9* illustrates the richness of regulatory mechanisms that control the postsynaptic levels of AMPARs and NMDARs via presynaptic expression of neurexins and cerebellins. In our studies, the changes in synaptic transmission induced by disrupting neurexin-cerebellin signaling are large, resulting in major alterations in the information processing of any circuit containing affected synapses. Since both cerebellin expression (*Hrvatin et al., 2018*; *Ibata et al., 2019*) and neurexin alternative splicing at SS4 (*Iijima et al., 2011*; *Ding et al., 2017*; *Flaherty et al., 2019*) may be activity-dependent, the unexpected signaling mechanism we describe likely also mediates activity-dependent plasticity. Thus, activity-dependent gene expression changes in a pre- or postsynaptic neuron may regulate the AMPAR- and NMDAR-composition via Nrxn1$^{SS4+}$/Nrxn3$^{SS4+}$→Cbln signaling. This type of AMPAR and NMDAR plasticity suggests a novel mechanism of circuit plasticity that may contribute to fundamental brain functions such as learning and memory (*Silver, 2010*; *Josselyn and Tonegawa, 2020*).

Needless to say, our findings raise major new questions. The current data at best are the beginning of an understanding of how neurexin-cerebellin signaling shapes synapses. Among major questions, it is puzzling why Nrxn3$^{SS4+}$ has no effect on mPFC synapses. Nrxn3$^{SS4+}$ is expressed in the mPFC but doesn't regulate AMPARs, suggesting it has a different function that is independent of Cbln2. In contrast, it is easier to understand why Nrxn1$^{SS4+}$ doesn't regulate NMDARs at parallel-fiber synapses since these synapses lack functional NMDARs (*Llano et al., 1991*; *Perkel et al., 1990*), but this finding also raises the question whether Nrxn1$^{SS4+}$ has another currently unknown function at these synapses. Neurexins can likely operate at the same synapses via binding to different ligands (*Seigneur et al., 2021*), a fascinating amplification of their functions that may also apply to parallel-fiber synapses. A further question is how the function of Cbln1 and Nrxn3$^{SS4+}$ in regulating AMPARs relates to the well-described parallel-fiber synapse loss in constitutive Cbln1 KO mice. It is possible that the prolonged Ca$^{2+}$ overload driven by long-lasting synaptic activity which is caused by overactivation of AMPARs leads to synaptotoxicity (*Green, 2009*; *Finch et al., 2012*), thereby harming parallel-fiber synapses. Another plausible explanation could be that at cerebellar parallel-fiber synapses, Cbln1 has additional functions that are not operative for cerebellins in subiculum synapses. Finally, how exactly neurexin-cerebellin signals are transduced postsynaptically via GluD's, and how synapse specificity is achieved here, constitutes another challenging but important question. Future studies will have to explore these intriguing questions.

In summary, our data spanning diverse genetic manipulations in multiple brain regions establish a general function for Cbln1 and Cbln2 in the trans-synaptic regulation of NMDARs and AMPARs that is regulated by presynaptic Nrxn1$^{SS4+}$ and Nrxn3$^{SS4+}$, respectively. Remarkably, this signaling pathway differentially operates in different neural circuits, creating a panoply of synaptic regulatory

mechanisms that are inherently plastic and enhance the activity-dependent capacity for information processing by neural circuits.

# Materials and methods

**Key resources table**

| Reagent type (species) or resource | Designation | Source or reference | Identifiers | Additional information |
|---|---|---|---|---|
| Antibody | Anti-vGluT1 | Millipore | Cat. No. AB5905 | 1:1000 |
| Antibody | Anti-Homer1 | Millipore | Cat. No. ABN37 | 1:1000 |
| Antibody | Anti-GAD65 | DSHB | Cat. No. mGAD6-a | 1:500 |
| Antibody | Anti-MAP2 | Millipore | Cat. No. AB5622 | 1:1000 |
| Antibody | Anti-Synaptotagmin 1 | Südhof lab | CL41.1 | 1:1000 |
| Antibody | Anti-Neurexin | Südhof lab | G394 | 1:500 |
| Antibody | Anti-CASK | BD Transduction Laboratories | Cat. No. 610782 | 1:1000 |
| Antibody | Anti-PSD95 | Südhof lab | L667 | 1:500 |
| Antibody | Anti-Synapsin | Südhof lab | E028 | 1:1000 |
| Antibody | Anti-Neuroligin-1 | Südhof lab | 4F9 | 1:500 |
| Antibody | Anti-β-actin | Sigma | Cat. No. A1978 | 1:10000 |
| Antibody | Anti-Calbindin | Sigma | Cat. No. C9848 | 1:2000 |
| Sequence-based reagent | Cbln1 in-situ probe | Advanced Cell Diagnostics | Cat. No. 538491-C2 | |
| Sequence-based reagent | Cbln2 in-situ probe | Advanced Cell Diagnostics | Cat. No. 428551 | |
| Recombinant DNA reagent | Lenti-hSyn-Cre-eGFP | *Aoto et al., 2013* | N/A | Lentiviral construct to express Cre and eGFP |
| Recombinant DNA reagent | Lenti-hSyn-eGFP | *Aoto et al., 2013* | N/A | Lentiviral construct to express eGFP |
| Recombinant DNA reagent | Lenti-CAG-Cre-eGFP | This paper | N/A | Lentiviral construct to express Cre and eGFP |
| Recombinant DNA reagent | Lenti-CAG-eGFP | This paper | N/A | Lentiviral construct to express eGFP |
| Recombinant DNA reagent | pAAV-hSyn-Cre-eGFP | *Aoto et al., 2015* | N/A | AAV construct to express Cre and eGFP |
| Recombinant DNA reagent | pAAV-hSyn-eGFP | *Aoto et al., 2015* | N/A | AAV construct to express eGFP |
| Recombinant DNA reagent | pAAV-hSyn-eGFP-p2A-Nrxn1βSS4+/- | *Dai et al., 2019* | N/A | AAV construct to express eGFP and Nrxn1βSS4+/- |
| Recombinant DNA reagent | pAAV-hSyn-eGFP-p2A-Nrxn3βSS4+/- | *Dai et al., 2019* | N/A | AAV construct to express eGFP and Nrxn3βSS4+/- |
| Chemical compound, drug | CNQX | Tocris | Cat. No. 0190 | |
| Chemical compound, drug | Picrotoxin | Tocris | Cat. No. 1128 | |
| Chemical compound, drug | TTX | Fisher Scientific | Cat. No. 50-753-2807 | |
| Chemical compound, drug | Biocytin | Sigma | Cat. No. B4261 | |

*Continued on next page*

*Continued*

| Reagent type (species) or resource | Designation | Source or reference | Identifiers | Additional information |
|---|---|---|---|---|
| Chemical compound, drug | Streptavidin Alexa 647 | Thermo Fisher | Cat. No. S32354 | 1:1000 |
| Genetic reagent (*Mus musculus*) | Mouse: C57BL/6J wildtype | The Jackson Laboratory | Jax Stock no: 000664 | |
| Genetic reagent (*Mus musculus*) | Mouse: Nrxn1-SS4+, Nrxn3-SS4+cKI | *Dai et al., 2019*; *Aoto et al., 2013* | N/A | |
| Genetic reagent (*Mus musculus*) | Mouse: Cbln1, Cbln2, Cbln12 cKO, Cbln2 KO | *Seigneur and Südhof, 2017* | N/A | |
| Software, algorithm | Clampfit 10 | Molecular Devices | https://www.moleculardevices.com/products/axon-patch-clamp-system/acquisition-and-analysis-software/pclamp-software-suite | |
| Software, algorithm | Igor software | Wavemetrics | https://www.wavemetrics.com/downloads | |
| Software, algorithm | Image Studio | LI-COR Biosciences | https://www.licor.com/bio/image-studio/ | |
| Software, algorithm | NIS-Elements AR Analysis | Nikon | https://www.microscope.healthcare.nikon.com/products/software/nis-elements/nis-elements-advanced-research | |
| Software, algorithm | Viewer III | Bioserve | http://www.biobserve.com/behavioralresearch/products/viewer/ | |
| Software, algorithm | Prism | GraphPad Software | https://www.graphpad.com/scientific-software/prism/ | |
| Software, algorithm | SigmaPlot | Systat Software | https://systatsoftware.com/sp/download.html | |

## Mice

The Cbln1 conditional KO and Cbln2 conditional/constitutive KO mouse lines were described in *Seigneur and Südhof, 2017*. SS4 +conditional knockin (cKI) mice of *Nrxn1* and *Nrxn3* were described previously (*Aoto et al., 2013*; *Dai et al., 2019*; *Dai et al., 2021*). All mice above were maintained on a mixed C57BL/6/SV129/CD1 (wild type) background. Primers (IDT) are used for genotyping are as follows: Nrxn1-SS4+, forward: 5'-AGACAGACCCGAACAACCAA-3', reverse: 5'-TGCTAGGCCTAT TTCAGATGCT-3'; Nrxn3-SS4+, forward: 5'-CTCCAACCTGTCATTCAAGGG-3', reverse: 5'-CTAC GGGCCGGTTATATTTG-3'; Cbln1, LoxP forward: 5'-TAGGG TGGACAGAGAAAAGG-'3, LoxP reverse: 5'- CTTCTAATCTGTCCTGACCACA-'3; Cbln2, LoxP forward: 5'-TAAAAGACAGTCCAGAGTTTTAGT C-3', LoxP reverse: 5'-TCAAATAGAGAGGAGTAAGCACA-3', and Recombined reverse: 5'-TTTC CTTGAAGGACTCCAATAG-3'. All mouse studies were performed according to protocols (#18846) approved by the Stanford University Administrative Panel on Laboratory Animal Care. In all studies, we examined littermate male or female mice.

## Single-molecule RNA FISH

As described in our previous study (*Dai et al., 2021*), P30 Wild type BL6 mice were euthanized with isofluorane and followed by transcardial perfusion with ice cold PBS. The brain were quickly dissected and embedded in OCT (Optimal Cutting Temperature) solution on dry ice. Horizontal sections with 16 µm thickness were cut by using Leica CM3050-S cryostat, mounted directly onto Superfrost Plus slides and stored in –80 °C until use. Single-molecule FISH for Cbln1 (Cat# 538491-C2) and Cbln2 (Cat# 428551) mRNA was performed using the multiplex RNAscope platform (Advanced Cell Diagnostics) according to manufacturer instructions. Fluorescent microscopy images were acquired at ×20 magnification using Olympus VS120 slide scanner.

## Semi-quantitative RT-PCR

For semi-quantitative RT-PCR measurements of neurexin SS4 alternative splicing (*Liakath-Ali and Südhof, 2021*), total RNA was extracted using TRIzol and cDNA was synthesized using the SuperScript

III First-Strand Synthesis System (Invitrogen) according to the manufacturer's instructions. PCR primers to detect Nrxn-SS4 isoforms (Forward, reverse): Nrxn1SS4, CTGGCCAGTTATCGAACGCT, GCGATGTT GGCATCGTTCTC; Nrxn2SS4, CAACGAGAGGTACCCGGC, TACTAGCCGTAGGTGGCCTT; Nrxn3SS4, ACACTTCAGGTGGACAACTG, AGTTGACCTTGGAAGAGACG; β-actin, TTGTTACCAACTGGGACGAC A, TCGAAGTCTAGAGCAACATAGC.

## mRNA measurements

mRNA was prepared from brain tissue directed from the subiculum or PFC region of P35-42 mice. RNA extraction was taken by using Trizol (Thermo Fisher, 15596026) and quantified using an ND-1000 spectrophotometer (NanoDrop, ThermoScientific). Quantitative RT-PCR was performed using the TaqMan Fast Virus One-Step Master Mix (Life Technologies) based on the manufacturer's instructions, and reactions were carried out and quantified using a QuantStudio 3 instrument (Applied Biosystems). Expression levels were normalized to β-actin (Applied Biosystems; cat. no. 4352933) as endogenous internal control. The following PrimeTime qPCR Assays (IDT) were used (shown as gene, primer1, probe, primer2 or predesigned): Nrxn1, ACTACATCAGTAACTCAGCACAG, CTTCTCCT TGACCACAGCCCCAT, ACAAGTGTCCGTTTCAAATCTTG; Nrxn3, TGCCACCTGAAATGTCTACC, CTACGACCACCACCCGAAAGAACC, ATCTGACGTGGGCTGAATG; Nrxn2, (Mm.PT.45.16500979); Cbln1 (Mm.PT.58.12172339); Cbln2 (Mm.PT.58.5608729); GluD1 (Mm.PT.58.32947175); GluD2 (Mm. PT.58.12083939).

## DNA constructs and viruses

hSyn-Cre-eGFP, hSyn-ΔCre-eGFP, CAG-Cre-eGFP, CAG-ΔCre-eGFP, full-length Nrxn1β$^{SS4+}$, Nrxn1β$^{SS4-}$, Nrxn3β$^{SS4+}$, and Nrxn3β$^{SS4-}$ were cloned into AAV-DJ vector (*Xu et al., 2012*; *Aoto et al., 2013*; *Dai et al., 2019*) for in vivo Cre-recombination or overexpression as previously described (*Dai et al., 2019*). The overexpression levels mediated by the viruses were quantified in microdissected brain tissue (please see details in *Dai et al., 2019*).

## Cell lines

HEK 293T cells were directly purchased from ATCC, which regularly validates cell lines. Cell lines were tested negative for mycoplasma contamination using the fluorochrome Hoechst DNA stain and the direct culture method.

## Slice electrophysiology

As previously described, electrophysiological recordings from acute hippocampal slices (*Dai et al., 2019*; *Dai et al., 2021*) or PFC (*Xu et al., 2012*) or cerebellum (*Zhang et al., 2015*) were essentially performed. In brief, slices were prepared from Cbln2$^{+/+}$ and Cbln2$^{-/-}$ mice at P35-42, or from all other mice at 2–3 weeks after stereotactic infection of AAVs (encode Cre, ΔCre, and various β-neurexins). Horizontal hippocampal slices (300 µm thickness) and Coronal PFC slices (250 µm thickness) were cut in a high sucrose cutting solution containing (in mM) 85 NaCl, 75 sucrose, 2.5 KCl, 1.3 NaH$_2$PO$_4$, 24 NaHCO$_3$, 0.5 CaCl$_2$, 4 MgCl$_2$ and 25 D-glucose. Sagittal cerebellum slices were sectioned in a low calcium solution containing (in mM) 125 mM NaCl, 2.5 mM KCl, 3 mM MgCl$_2$, 0.1 CaCl$_2$, 1.25 NaH$_2$PO$_4$, 25 NaHCO$_3$, 3 mM myo-inositol, 2 mM Na-pyruvate, 0.4 mM ascorbic acid, and 25 D-glucose. Slices were equilibrated in ACSF at 31 °C for 30 min, followed by room temperature for an hour. Hippocampal or PFC Slices were then transferred to a recording chamber containing ACSF solution maintained at 30.5 °C (in mM): 120 NaCl, 2.5 KCl, 1 NaH$_2$PO$_4$, 26.2 NaHCO$_3$, 2.5 CaCl$_2$, 1.3 MgSO$_4$-7 H$_2$O, 11 D-Glucose,~290 mOsm. Cerebellum slices were then transferred to a recording chamber containing ACSF solution maintained at 30.5 °C (in mM): 125 mM NaCl, 2.5 mM KCl, 1 mM MgCl$_2$, 2 CaCl$_2$, 1.25 NaH$_2$PO$_4$, 25 NaHCO$_3$, 3 mM myo-inositol, 2 mM Na-pyruvate, 0.4 mM ascorbic acid, and 25 D-glucose. To induce evoked synaptic responses in subiculum, a nichrome stimulating electrode was placed at the most distal portion of hippocampal CA1 region as shown in our previous studies (*Dai et al., 2019*; *Dai et al., 2021*). The firing type of subiculum neurons (burst-spiking vs. regular-spiking) was identified by injecting a depolarizing current immediately after breaking in and monitoring action potential patterns in current-clamp mode as previously described (*Aoto et al., 2013*; *Dai et al., 2019*). To induce evoked synaptic responses in mPFC, the electrode was placed at the border of L1 and L2/3 layer as illustrated in *Figure 6B* and the L5/6 layer pyramidal neurons were recorded (*Fénelon et al.,*

*2011*). To induce evoked synaptic responses in cerebellum, the electrode was placed at the parallel fibers in the distal molecular layer as illustrated in *Figure 8B* and the purkinje neurons were recorded (*Zhang et al., 2015*). AMPAR-EPSCs input/output curves, AMPAR/NMDAR ratios, NMDAR input/output curves, LTP, and mEPSCs (holding potentials = –70 mV for AMPAR-EPSCs,+40 mV for NMDAR-EPSCs, and +60 mV for NMDAR mEPSCs) were recorded with an internal solution containing (in mM): 117 Cs-methanesulfonate, 15 CsCl, 8 NaCl, 10 TEA-Cl, 0.2 EGTA, 4 Na2-ATP, 0.3 Na2-GTP, 10 HEPES, pH 7.3 with CsOH (~300 mOsm). All recordings were performed in the presence of 50 µM picrotoxin for AMPAR-EPSCs, 50 µM picrotoxin and 10 µM CNQX for NMDAR-EPSCs, and 50 µM picrotoxin and 0.5 µM TTX for mEPSCs. Paired-pulse ratios were monitored with interstimulus intervals of 20–2000 ms. LTP was induced by four tetani of 100 Hz stimulus trains applied for 1 s with 10 s intervals under voltage-clamp mode (holding potential = 0 mV). Pre-LTP (averaging last 5 mins as baseline) and post-LTP (averaging the last 5 mins) were recorded at 0.1 Hz. Paired-pulse ratios were measured with 40ms interstimulus intervals before and after LTP. Measurements of the AMPAR/NMDAR ratios were performed in 50 µM picrotoxin at holding potentials of –70 mV (AMPAR-EPSCs) or +40 mV (NMDAR-EPSCs, quantified at 50ms after the stimulus). All slopes of input/output ratio were calculated from 10 to 50 µA of input current except the cerebellum that was calculated from 10 to 100 µA of input current. All data were analyzed with the Igor software (WaveMetrics). Miniature events were hand-picked with a threshold of 5 pA by using the Igor software (*Dai et al., 2015*).

## Stereotactic Injections

Stereotactic injections of AAV into mice at P21 were performed essentially as described (*Xu et al., 2012*; *Aoto et al., 2013*; *Dai et al., 2019*; *Dai et al., 2021*). Briefly, P21 mice were anesthetized with Avertin, and viruses were injected using a stereotactic instrument (David Kopf) and a syringe pump (Harvard Apparatus) with ~0.85 µl of concentrated virus solution ($10^{8-9}$ TU) at a slow rate (0.1 l/min) iµnto the CA1 region of the intermediate hippocampus (Bregma coordinates (mm): AP: –3.1, ML:±3.4, DV: –2.5) or with ~0.4 µl of virus into subiculum region (Bregma coordinates (mm): AP: –3.3, ML:±3.3, DV: –2.5) or with ~0.8 µl of virus into mPFC region (Bregma coordinates (mm): AP:+1.25, ML:±0.3, DV: –1.0 mm and –1.5 mm received both 0.4 µl of virus) or with ~0.8 µl of virus into cerebellum lobe4-5 region (Bregma coordinates (mm): AP: –6.35, ML:±0.5, DV: –1.5 mm received both 0.4 µl of virus). After infection, viral mediated expression was confirmed by the presence of eGFP. Images (*Figures 4F, 6B and 7B*) were taken using a Nikon confocal microscope (A1Rsi) with a 10 x objective (PlanApo, NA1.4) with 1024x1024 pixel resolution. The fluorescence of all slices prepared for physiology was confirmed under a fluorescence microscope (Olympus).

## Immunohistochemistry

For hippocampal cryosections were performed as described (*Dai et al., 2019*; *Dai et al., 2021*). Briefly, mice were anesthetized with isoflurane and perfused with 10 ml PBS followed by 30 ml 4% PFA in 1 x PBS using a perfusion pump (2 ml/min). Whole brains were dissected out and kept in PFA for 6 hours, then post-fixed in 30% sucrose (in 1×PBS) for 24 h-48 h at 4 °C. Horizontal brain sections (30 µm) were collected at –20 °C with a cryostat (Leica CM1050). Sections were washed with PBS and incubated in blocking buffer (0.3% Triton X-100 and 5% goat serum in PBS) for 1 hr at RT, and incubated overnight at 4 °C with primary antibodies diluted in blocking buffer (anti-vGluT1, 1:1000, guinea pig, Millipore and anti-MAP2, 1:1000, rabbit, Millipore). Sections were washed three times for 10 min each in 1 x PBS, followed by treatment with secondary antibodies (1:1000, Alexa 405, Alexa 647) at 4 °C overnight, then washed three times for 10 min each with 1 x PBS. All incubations were performed with agitation. All sections were then mounted on superfrost slides and covered with Fluo-romount-G as previously described. Serial confocal z-stack images (1 µm step for 10 µm at 1024x1,024 pixel resolution) were acquired using a Nikon confocal microscope (A1Rsi) with a 60 x oil objective (PlanApo, NA1.4). All acquisition parameters were kept constant among different conditions within experiments. For data analysis (n≥3 animals per condition), maximum intensity projections were generated for each image, and average vGlut1 intensity (mean ± S.E.M) calculated from the entire area of subiculum (object size range 0.05–0.21 mm²). An example cerebellum slice was stained with vGluT1 (anti-vGluT1, 1:1000, guinea pig, Millipore) and Calbindin (anti-calbindin, 1:2000, mouse, Sigma). For double labeling of presynaptic marker vGlut1 (anti-vGluT1, 1:1000, guinea pig, Millipore) and postsynaptic maker Homer1 (anti-Homer1, 1:1000, rabbit, Milllipore), PFC coronal sections from constitutive

Cbln2 WT and KO are prepared exactly as described above. Then, we acquired images using a Nikon A1 Eclipse Ti confocal microscope with 100 x objective and 0.25 µm Z-stacks at 0.06 µm/pixel resolution, and nine sections were acquired and maximum pixel intensity projections were generated. For synaptic puncta quantification, images were thresholded by intensity to exclude background signals and the puncta size (0,1–3 µm$^2$) was quantified to calculate the mean intensity, size, and density.

## Immunoblotting

Immunoblotting was performed as described previously (*Seigneur and Südhof, 2018*; *Patzke et al., 2019*; *Dai et al., 2021*; *Patzke et al., 2021*). Briefly, dissected hippocampal tissue were homogenized in Laemmli buffer (12.5 mM Tris-HCl, pH 6.8, 5 mM EDTA, pH 6.8, 143 mM β-mercaptoethanol, 1% SDS, 0.01% bromophenol blue, 10% glycerol), boiled and separated by SDS–PAGE at 100 V for about 1.3 hr, then transferred onto nitrocellulose membranes using the Trans-Blot Turbo transfer system (Bio-Rad). Membranes were then blocked with 5% milk in TBS containing 0.1% Tween 20 (TBST) at RT for 1 hr, and then incubated in primary antibody overnight at 4 °C. Membranes were washed 3 X with TBST, then incubated in fluorescent labeled secondary antibodies (donkey anti-rabbit IR dye 680/800CW, 1:10000; donkey anti-mouse IR dye 680/800CW, 1:10,000; and donkey anti-guinea pig IR dye 680RD, 1:10,000; LI-COR Bioscience). Membranes were scanned using an Odyssey Infrared Imager and analyzed with the Odyssey software (LI-COR Biosciences). Intensity values for each protein were first normalized to actin then to the control sample. The antibodies used are as follows: anti-Neuroligin-1 mouse (1:500; Südhof lab; 4F9), anti-β-actin mouse (1:10000; Sigma-Aldrich; Cat# A1978), anti-PSD95 rabbit (1:500; Südhof lab; L667), anti-Synapsin rabbit (1:1000; Südhof lab; E028), anti-CASK mouse (1:1000; BD Transduction Laboratories; Cat# 610782), anti-Neurexin rabbit (1:500; Südhof lab; G394), anti-GAD65 mouse (1:500; DSHB; Cat# mGAD6-a), anti-Synaptotagmin-1 mouse (1:1000, Südhof lab; CL41.1), and anti-vGluT1 guinea pig (1:1000; Millipore; Cat# AB5905).

## Single-cell biocytin labeling

As described in previous study, whole-cell recordings with voltage clamp at –70 mV for about 10–15 mins. The cesium methanesulfonate internal solution was made as described above with 2 mg/mL Biocytin (Sigma Cat#B4261). Then slices were transferred to 4%PFA/PBS and fixed one hour in room temperature. Slices were washed 3x5 min with PBS, permeabilized in 0.3% Triton-X100/PBS for 30 min, and blocked in 5% normal goat serum (NGS)/0.1% Triton-X100/PBS at room temperature for 1 hr. Subsequently, slices were incubated in Streptavidin Alexa 647 (Invitrogen Cat#S32357) diluted 1:1000 in 5% NGS/0.1% Triton-X100/PBS at 4 °C overnight, washed 5x5 min with PBS and mounted with 0 thickness coverglass (Assistent Cat#01105209). Images were acquired using a Nikon A1 Eclipse Ti confocal microscope with 20 x and 100 x objectives, operated by NIS-Elements AR acquisition software. For spine imaging, Z-stacks were collected at 0.2 µm with 0.06 µm/pixel resolution, and 6–10 dendrites were analyzed per cell.

## Two-chamber avoidance test

Littermate Cbln2 WT and Cbln2 KO male mice were generated from crossing heterozygous Cbln2$^{+/-}$ mice. Mice were handled daily for 5 days prior to behavioral experiments starting at P45. Mice were maintained with a normal 12/12 hr daylight cycle, and analyzed in the assay sequence and at the time shown in *Figure 2A*. The modified protocol was performed as described previously (*Dai et al., 2019*) and was based on previous studies (*Ambrogi Lorenzini et al., 1984*; *Cimadevilla et al., 2001*; *Qiao et al., 2014*). Briefly, two chambers (left and right) were designed with different visual cues (*Figure 2B*) under dim light with a gate between them (Shuttle box, Med Associates Inc). The right chamber has a foot shock with electric current (intensity: 0.15 mA, duration: 2 s). Mice can explore both chambers freely. At the training day, mice will be put in left chamber. Once they go to the right chamber, they will get a foot shock after a 2 s delay. In this case, they will return back immediately to the left chamber. This is one trial of learning which is counted as one entry. It may come as another trial, once they visit right chamber again. This training process will be completed until mice are able to stay in left "safe" chamber more than 2 min. After 1 day and 7 days, they will be tested by putting back into left chamber to record latency to enter the right chamber and the number of entries in 2 min. Using this approach, two groups of Cbln2 WT and KO mice were tested. All behavior assays were carried out and analyzed by researchers blindly.

## Quantification and statistical analysis

All data are shown as means ± SEMs, with statistical significance (*=$p < 0.05$, **=$p < 0.01$ and ***=$p < 0.001$) determined by Student's t-test or two-way analysis of variance (ANOVA). Non-significant results ($p > 0.05$) are not specifically identified.

## Materials availability

All reagents produced in this study, including recombinant DNA plasmids and mouse lines, are openly distributed to the scientific community and freely shared upon request.

## Acknowledgements

This study was supported by a grant from the NIMH (MH052804 to TCS) and fellowships to KL-A from the European Molecular Biology Organization (ALTF 803–2017) and the Larry L Hillblom Foundation (2020-A-016-FEL).

## Additional information

### Funding

| Funder | Grant reference number | Author |
| --- | --- | --- |
| National Institute of Mental Health | MH052804 | Thomas C Südhof |
| European Molecular Biology Organization | ALTF 803-2017 | Kif Liakath-Ali |
| Larry L. Hillblom Foundation | 2020-A-016-FEL | Kif Liakath-Ali |

The funders had no role in study design, data collection and interpretation, or the decision to submit the work for publication.

### Author contributions

Jinye Dai, Conceptualization, Resources, Data curation, Software, Formal analysis, Supervision, Validation, Investigation, Visualization, Methodology, Writing – original draft, Project administration, Writing – review and editing; Kif Liakath-Ali, Resources, Data curation, Formal analysis, Funding acquisition, Investigation, Visualization, Writing – review and editing; Samantha Rose Golf, Formal analysis, Investigation, Writing – review and editing; Thomas C Südhof, Conceptualization, Resources, Data curation, Supervision, Funding acquisition, Validation, Investigation, Visualization, Methodology, Writing – original draft, Project administration, Writing – review and editing

### Author ORCIDs

Jinye Dai http://orcid.org/0000-0002-8497-3154
Kif Liakath-Ali http://orcid.org/0000-0001-9047-7424
Thomas C Südhof http://orcid.org/0000-0003-3361-9275

### Ethics

All mouse studies were performed according to protocols (#18846) approved by the Stanford University Administrative Panel on Laboratory Animal Care.

### Decision letter and Author response

Decision letter https://doi.org/10.7554/eLife.78649.sa1
Author response https://doi.org/10.7554/eLife.78649.sa2

## Additional files

### Supplementary files

• MDAR checklist

## Data availability

All numerical data and P values within this study have been included in the manuscript.

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
