## [Editor Report]

This manuscript examines signaling complexes involving neurexins and cerebellins that bridge the two sides of the synaptic junction. Through carefully executed experiments, the study shows that the basic framework of the complexes operates broadly across different synapses in the brain albeit with subtle differences. This work is of broad interest to neuroscientists studying mechanisms regulating synapse formation and maintenance.

---

## [Decision Letter]

**Decision letter after peer review:**

Thank you for submitting your article "Distinct Neurexin-Cerebellin Complexes Control AMPA-and NMDA-Receptor Responses in a Circuit-Dependent Manner" for consideration by *eLife*. Your article has been reviewed by 3 peer reviewers, one of whom is a member of our Board of Reviewing Editors, and the evaluation has been overseen by Gary Westbrook as the Senior Editor. The reviewers have opted to remain anonymous. The reviewers have discussed their reviews with one another, and the Reviewing Editor has drafted this to help you prepare a revised submission.

Essential revisions:

This study is a follow-up to a series of elegant work by the authors (Aoto et al., 2013; Dai et al., Neuron 2019; Dai et al., Nature 2021) in establishing a surprising role for the presynaptic adhesion molecules, neurexin (Nrxn) containing the SS4+ splice site, in differentially controlling postsynaptic NMDA and AMPA receptors by forming links through a shared system of extracellular cerebellins (Cbln) and postsynaptic GluD. Here the two major findings are reported: (1) the previously described function of Nrxn-Cbln-GluD complexes at mature CA1 to subiculum synapses extend to different synapse types in the subiculum and also to some synapses in the cortex and the cerebellum and (2) the Nrxn-Cbln-GluD complexes do not appear to play a role in synapse formation as proposed by others. The experiments have been expertly executed. However, the scope of the present findings is limited over the previous work. Moreover, one of the key conclusions – of the lack of involvement of Nrxn-Cbln-GluD in synapse formation – could be further strengthened with additional data. To address these concerns, the following essential revisions are requested. The full reviews are appended below, which will help clarify the concerns of the individual reviewers. The authors should address also all the points raised in the individual reviews, particularly those involving re-analysis of data and text edits for clarifications/explanations.

1) To clarify the role of the Nrx-Cbln-GluD complex in controlling synapse formation/development, either a detailed and quantitative synaptic analysis of both pre and postsynaptic markers and/or quantification of dendritic spine density from individual cells should be performed in control WT neurons and neurons deficient for the trans-synaptic adhesion complex.

2) To investigate whether AMPAR and NMDAR changes result from changes in the number of active synapses and/or quantal size, the authors must perform a detailed quantal analysis of electrophysiological recordings.

3) To validate the specificity of their approach, the authors must investigate through immunoblotting and/or RT-qPCR whether the deletion of Cbln1 and/or Cbln2 affect the expression of other Cblns isoforms (Cbln2 and/or Cbln4) as well as Nrxns1/3 and GluD1/2 to ensure that there is no compensatory effect arising from the genetic deletion of Cbln1/2.

4) L272- 'Cbln1 is also expressed in the subiculum, albeit at lower levels'. The images from figure S3 show no Cbln1 mRNA signal (in agreement with Otsuka et al., 2016). The authors should change the image and/or provide some quantitative analysis to support this conclusion.

*Reviewer #1 (Recommendations for the authors):*

Figure 1. Could the authors rule out the possibility that in regular spiking cells, the block of LTP in Clbn2 KO cells is due to occlusion/saturation by the already high synaptic levels of AMPAR? Is LTD unaffected? If so, could one rescue LTP after inducing LTD? Alternatively, if LTP deficit represents a general defect in AMPAR trafficking as suggested, then LTD might be also compromised. It would be informative to clarify these points.

Figure 3. Since the role for Clbns in synapse formation is a contentious issue, the authors should compare synapse density in an input-specific manner. An overall lack of change in VGluT1 staining intensity in hippocampus is certainly one measure of synapse density, although it may mask changes at the level of different synapse types. Filling the identified neurons and assessing spine density along with posthoc labelling with a presynaptic marker is expected to give a more definitive measure.

Figure 5. It would be much more compelling if the Cbln1/2 DKO could be directly compared also to Cbln1 KO or Cbln2 KO to demonstrate that the extent changes observed are comparable between single and double KOs. Moreover, the quantification of NMDAR mEPSCs as shown, is not convincing given the high levels of baseline noise, which exceed the amplitude of the tail responses as seen in the raw traces. Although indirect, perhaps measurements of NMDAR-dependent calcium signals could provide a clearer dataset.

*Reviewer #3 (Recommendations for the authors):*

(1) To address further the role of the Nrx-Cbln-GluD pathway in the control of synaptic formation/development, the authors must perform a detailed and quantitative synaptic analysis of both pre and postsynaptic markers and quantify the density, intensity and apposition of corresponding clusters in the three models (pyramidal neurons from subiculum and mPFC and cerebellar Purkinje cells). To compare with other studies (e.g., Tao et al., 2018; Shibata et al., 2021), they should also quantify dendritic spine density from individual cells.

(2) To investigate whether AMPAR and NMDAR changes result from changes in the number of active synapses and/or quantal size, the authors must perform a detailed quantal analysis of electrophysiological recordings and/or provide quantitative measurements of AMPARs and NMDARs at the synaptic level (e.g., through immunostainings or biochemical analysis of synaptosomal fractions).

(3) To validate the specificity of their approach, the authors must investigate through immunoblotting and/or RT-qPCR whether the deletion of Cbln1 and/or Cbln2 affect the expression of other Cblns isoforms (Cbln2 and/or Cbln4) as well as Nrxns1/3 and GluD1/2 to ensure that there is no compensatory effect arising from the genetic deletion of Cbln1/2.

(4) The authors should discuss further their biological model and propose a molecular mechanism by which the Nrxn-Cbln-GluD pathway could directly up-regulate AMPARs or down-regulate NMDARs (depending on the Nrxn and Cblns engaged at each of the connection), without altering synapse formation.

(5) L94 – 95- 'Parallel-fiber synapses develop initially normally, but are subsequently lost (Kashiwabuchi et al., 1995; Kurihara et al., 1997; Hirai et al., 2005; Takeuchi et al., 2005)'. This statement is incorrect. The authors are right when saying that the initial parallel fiber synapses develop initially normally, i.e., during the first 10 postnatal days (Kurihara et al., 1997). However, this phase generates only a small fraction of the total number of PF->PC synapses. Actually, most PF-PCs synapses assemble in the following 2 weeks (e.g., Altman et al., J Comp Neurol 1972; Takacs and Hamori, J Neurosci Research, 1994), i.e, when the GluD2 KO mice start displaying abnormal PF synapse number (Kurihara et al., 1997). The statement that PF synapses develop initially normally but are subsequently lost is therefore incorrect.

(6) L114 – 'an RNAi-induced suppression of Cbln2 expression was found to suppress formation of excitatory synapse numbers in the CA1 region of the hippocampus'. This citation is not correct as Tao et al., performed RNAi-mediated GluD1 knock-down +/- Cbln2 overexpression.

(7) L114-L118 – The authors compare studies that used different readouts to monitor synapse number. While Tao et al., quantified spine number and AMPAR / NMDAR-EPSCs in 6d organotypic slices or acute slices from 1 month-old mice, Seigneur and Südhof, as in the present study, quantified average vGluT1 staining intensity from histological sections. The comparison is thus limited and the different methods to quantify synapses should be mentioned.

(8) L121-L122 – 'which is also puzzling since Cbln4 does not bind to GluD1 (Zhong et al., 2017; Cheng et al., 2016)'. Both studies tested the binding of Cblns to GluD2, not GluD1. Moreover, Cheng et al., only looked at Cbln1, not Cbln4. Therefore, the citations are not accurate.

(9) L272 – 'Cbln1 is also expressed in the subiculum, albeit at lower levels'. The images from figure S3 show no Cbln1 mRNA signal (in agreement with Otsuka et al., 2016). The authors should change the image and/or provide some quantitative analysis to support this conclusion.

(10) Figure 6C – It is not clear how AMPAR-EPSCs and NMDAR-EPSCs were normalized across recordings for experiments performed in the mPFC since no input-output curve was performed.

(11) L492 – the argument that 'overactivation of AMPARs may lead to synaptotoxicity thereby explaining synapse loss' is interesting but not supported by any data in the paper nor by relevant citations. It should therefore be discussed further. Actually, the decrease in NMDARs is not in favor of the synaptotoxicity hypothesis.

(12) The discrepancy between the current study and previous reports needs to be further discussed. In particular, Tao et al., reported that Cbln2 overexpression enhances AMPAR and NMDAR EPSCs amplitudes while the present study shows that Cbln2 KO enhances AMPAR EPSCs while decreasing NMDAR-current. The surprising similar effect of Cbln2 overexpression and KO on AMPARs-EPSCs needs to be addressed.

[Editors’ note: further revisions were suggested prior to acceptance, as described below.]

Thank you for resubmitting your work entitled "Distinct Neurexin-Cerebellin Complexes Control AMPA- and NMDA-Receptor Responses in a Circuit-Dependent Manner" for further consideration by *eLife*. Your revised article has been evaluated by Gary Westbrook (Senior Editor) and a Reviewing Editor. The manuscript has been improved but please fully address the concerns raised by Reviewer #3 by editing the text.

*Reviewer #3 (Recommendations for the authors):*

In the revised manuscript, the authors have made several improvements and addressed many of my criticisms. However, I still have some concerns about the statements made in the introduction regarding the cerebellum.

L96-99: "parallel fiber synapses develop initially, but subsequently decline, with a 40-50% decrease in parallel-fiber synapses in adult Cbln1 or GluD2 KO mice". This sentence is misleading as it suggests that PF synapses form normally and that the decreased number entirely results from abnormal elimination. Actually, as acknowledged by the authors, the current data do not allow to exclude the possibility that the GluD2 KO impairs the critical phase of PF synapse formation which occurs during the 2nd and 3rd weeks. Therefore, this claim should be toned down.

L137: "Even in the cerebellum of Cbln1 KO mice, the observed synapse loss is not accompanied by an equivalent decrease in spine density (Hirai et al., 2005)". In contrast to other circuits like PFC, it is well established that PC spines spontaneously form and can remain 'free' even in the absence of PF inputs. In this regard, PC spines cannot be used as a proxy for synapse number and a direct comparison with PFC is not appropriate.

Overall, while data from the authors do support that Nrx-Cbln-GluD signaling is not required for synapse formation in PFC and subiculum (which does not mean that they do not play any role), I think that the idea that the impact of Cbln / GluD2 deletion on PF synapse number in PCs primarily results from increased elimination rather than defect in synapse formation is only weakly supported by published data and appears overstated through the manuscript.

---

## [Author Response]

Essential revisions:This study is a follow-up to a series of elegant work by the authors (Aoto et al., 2013; Dai et al., Neuron 2019; Dai et al., Nature 2021) in establishing a surprising role for the presynaptic adhesion molecules, neurexin (Nrxn) containing the SS4+ splice site, in differentially controlling postsynaptic NMDA and AMPA receptors by forming links through a shared system of extracellular cerebellins (Cbln) and postsynaptic GluD. Here the two major findings are reported: (1) the previously described function of Nrxn-Cbln-GluD complexes at mature CA1 to subiculum synapses extend to different synapse types in the subiculum and also to some synapses in the cortex and the cerebellum and (2) the Nrxn-Cbln-GluD complexes do not appear to play a role in synapse formation as proposed by others. The experiments have been expertly executed. However, the scope of the present findings is limited over the previous work. Moreover, one of the key conclusions – of the lack of involvement of Nrxn-Cbln-GluD in synapse formation – could be further strengthened with additional data. To address these concerns, the following essential revisions are requested. The full reviews are appended below, which will help clarify the concerns of the individual reviewers. The authors should address also all the points raised in the individual reviews, particularly those involving re-analysis of data and text edits for clarifications/explanations.

We appreciate the editors’ and reviewers’ positive assessment of our study, which we believe significantly shapes our views of the role of cerebellins in synaptic signaling.

1) To clarify the role of the Nrx-Cbln-GluD complex in controlling synapse formation/development, either a detailed and quantitative synaptic analysis of both pre and postsynaptic markers and/or quantification of dendritic spine density from individual cells should be performed in control WT neurons and neurons deficient for the trans-synaptic adhesion complex.

We have performed extensive additional experiments to address this point. We concur that although we reported immunocytochemistry experiments for the presynaptic maker vGlut1 in the subiculum to quantify synaptic density in the original paper, this approach alone might be considered inconclusive. To address this concern and, at the same time, to generate more value for the field, we decided to expand the scope of our project and to perform further experiments along the lines suggested by the reviewers.

Instead of the hippocampus, however, we opted to perform these experiments in the prefrontal cortex for two reasons. First, we already previously presented extensive data in the subiculum showing that various neurexin, cerebellin, and GluD deletions have no effect on synapses numbers. Doing more of the same seemed a bit superfluous, but showing this for a different brain regions is very informative. Second, a recent well-noted *Nature* paper claimed that *Cbln2* controlled spine numbers in the prefrontal cortex, and thus regulated synapse numbers (Shibata et al., “Hominini-specific regulation of *CBLN2* increases prefrontal spinogenesis”). Notably, this paper’s conclusion that *Cbln2* controls spinogenesis has large implications not only for human evolution, but also for our understanding of the role of *Cbln2*-associated synaptic adhesion in constructing neural circuits. In addition, the *Nature* paper suggested that *Cbln2* controls spinogenesis in mouse prefrontal cortex, and thus are incongruent with our data. Because of this situation, we decided to test this question directly in prefrontal cortex.

Following the reviewers’ and editors’ suggestions, we measured spine numbers as well as synapse density using co-labeling of synapses for pre- and postsynaptic markers as an approach. The experiments were carried out in littermate constitutive *Cbln2* KO and control mice in order to ensure that we did not miss a possible developmental phenotype, since in the constitutive *Cbln2* KO mice, *Cbln2* is absent throughout the life of a mouse. The new results show unequivocally that the *Cbln2* KO causes no change in synapse density or in spine numbers (new Figure 7). This is consistent with our current and previous results in the hippocampus.

Why are our results different from those of Shibata et al.? Their *Nature* paper did not explore the effects of a direct genetic manipulation – it used a ‘humanization’ of a putative enhancer for the *Cbln2* gene that causes a small increase in *Cbln2* expression (~25%), but a robust increase (40-50%) in dendritic spines. No actual test of the role of *Cbln2* in regulating spine numbers was performed, and the effect of the genetic manipulation on spine numbers was disproportionately high compared to the effect on *Cbln2* expression. Thus, a plausible explanation for the discrepancy between our results and the conclusions of Shibata et al., (2021) may be that the enhancer manipulation could have also affected other genes that might drive spine formation.

2) To investigate whether AMPAR and NMDAR changes result from changes in the number of active synapses and/or quantal size, the authors must perform a detailed quantal analysis of electrophysiological recordings.

We respectfully beg to differ from this recommendation for two reasons.

First, a true quantal analysis of a synapse is a major project on its own, traditionally published as a stand-alone paper. Rigorous quantal analyses are difficult to perform on synapses that cannot be readily monitored by paired recordings or stimulation of single nerve terminals. To the best of our knowledge, no detailed quantal analysis has ever been done for CA1→subiculum synapses, and may not even be possible given the convergent/divergent synaptic input/output relations. Adding this to the current paper, if feasible, would mean at least 6-12 months of extra work.

Second, a detailed quantal analysis is unnecessary in our study, and is “overkill”. We show that synapse numbers don’t change, as now extensively documented in additional experiments. Our data also demonstrate that AMPAR and NMDAR changes are in the opposite directions, which cannot possibly be explained by a change in the number of active synapses. Moreover, the mEPSC frequency increases, which is incompatible with the decrease in synapse numbers that would have been expected by alternative explanations. Furthermore, we performed in the current paper a detailed analysis of miniature quantal size of AMPA mEPSCs or NMDA mEPSCs, showing that there is no change. In addition, we previously demonstrated that the opposing changes in NMDAR and AMPAR responses are caused by changes in surface receptor responses (Dai et al., 2019). Finally, we have now added data demonstrating further that the release probability doesn’t change by calculating the coefficient of variation. Based on the entirety of these findings, our data conclusively rule out the possibility of a decrease in synapse numbers or in the percentage of active synapses – neither is consistent with the data. A detailed quantal analysis would simply confirm this conclusion.

3) To validate the specificity of their approach, the authors must investigate through immunoblotting and/or RT-qPCR whether the deletion of Cbln1 and/or Cbln2 affect the expression of other Cblns isoforms (Cbln2 and/or Cbln4) as well as Nrxns1/3 and GluD1/2 to ensure that there is no compensatory effect arising from the genetic deletion of Cbln1/2.

We agree, and have performed the suggested experiments. The new results are now shown in the new Figure 5—figure supplement 2, fully confirming the reviewers’ predictions.

4) L272- 'Cbln1 is also expressed in the subiculum, albeit at lower levels'. The images from figure S3 show no Cbln1 mRNA signal (in agreement with Otsuka et al., 2016). The authors should change the image and/or provide some quantitative analysis to support this conclusion.

We agree that this issue needed to be presented better, and we have changed the image. The extremely high expression level of *Cbln1* in the cerebellum causes the *Cbln1* signal to become dim in other brain regions if all images are analyzed with the same settings. We have now increased the gain of the *Cbln1* in situ hybridization signal strength for other brain regions to demonstrate that *Cbln1* is actually expressed in these regions, albeit at much lower levels. This is noted in the revised legends. Additionally, to provide an overall quantitative analysis, we performed qRT-PCR measurements in both *Cbln2* WT and KO mice to determine the mRNA levels of *Cbln1* and *Cbln2* in the prefrontal cortex and subiculum. The results confirm that *Cbln2* has much higher expression level than *Cbln1* in the prefrontal cortex or subiculum, although *Cbln1* is clearly detectable (new Figure 5 figure supplement 2).

Reviewer #1 (Recommendations for the authors):Figure 1. Could the authors rule out the possibility that in regular spiking cells, the block of LTP in Clbn2 KO cells is due to occlusion/saturation by the already high synaptic levels of AMPAR? Is LTD unaffected? If so, could one rescue LTP after inducing LTD? Alternatively, if LTP deficit represents a general defect in AMPAR trafficking as suggested, then LTD might be also compromised. It would be informative to clarify these points.

We cannot rule out the possibility that the high synaptic levels of AMPARs in *Cbln2* KO neurons occlude LTP. Another possibility is that the loss of LTP is due to an induction impairment caused by the reduction in NMDARs, although we used a very strong LTP induction protocol (4 x 100 Hz for 1 s). We now note these possibilities in the paper, and did not mean to imply that an AMPAR trafficking impairment is the direct cause of the loss LTP, although for all alternative explanations AMPAR and NMDAR trafficking impairments would be at least indirect causes for the loss of LTP.

Figure 3. Since the role for Clbns in synapse formation is a contentious issue, the authors should compare synapse density in an input-specific manner. An overall lack of change in VGluT1 staining intensity in hippocampus is certainly one measure of synapse density, although it may mask changes at the level of different synapse types. Filling the identified neurons and assessing spine density along with posthoc labelling with a presynaptic marker is expected to give a more definitive measure.

We agree, and have performed the suggested experiment as described above (introductory point 1).

Figure 5. It would be much more compelling if the Cbln1/2 DKO could be directly compared also to Cbln1 KO or Cbln2 KO to demonstrate that the extent changes observed are comparable between single and double KOs. Moreover, the quantification of NMDAR mEPSCs as shown, is not convincing given the high levels of baseline noise, which exceed the amplitude of the tail responses as seen in the raw traces. Although indirect, perhaps measurements of NMDAR-dependent calcium signals could provide a clearer dataset.

We agree that it would have been best if we could directly compare *Cbln1*/2 DKO to *Cbln1* KO or *Cbln2* KO mice with the same genetic background in the same experiments. However, every mouse line has to be maintained separately for multiple generations, and it is very difficult and expensive to analyze littermates for all various genotypes. Moreover, the results from *Cbln2* conditional KO mice (Figure 1—figure supplement 2) and *Cbln1*/2 double conditional KO mice (Figure 5) are nearly identical, suggesting that there is no major genetic background effect.

We also agree that quantifications of NMDAR mEPSCs are questionable, and believe that we have presented these data in the paper with the appropriate caution. Our point here is that, different from AMPAR mEPSCs, NMDAR mEPSCs exhibit no increase. We used a holding potential of +60 mV even though the baseline noise is much higher because the higher holding potential increases the driving force and thereby enhances the NMDAR mEPSC amplitude. This enabled us to estimate NMDAR mEPSCs. After averaging many events, which reduced the baseline noise dramatically, we quantified the total charge using averaged mEPSC traces from different cells, not individual events. We concur that these data only allow relative conclusions between different genetic conditions, and are not useful as absolute measurements.

Reviewer #3 (Recommendations for the authors):(1) To address further the role of the Nrx-Cbln-GluD pathway in the control of synaptic formation/development, the authors must perform a detailed and quantitative synaptic analysis of both pre and postsynaptic markers and quantify the density, intensity and apposition of corresponding clusters in the three models (pyramidal neurons from subiculum and mPFC and cerebellar Purkinje cells). To compare with other studies (e.g., Tao et al., 2018; Shibata et al., 2021), they should also quantify dendritic spine density from individual cells.

We agree and have performed this experiment as described in our response to introductory comment 1. However, we feel that performing these labor-intensive experiments in all three synapses is a bit excessive, and focused on the prefrontal cortex given the recent *Nature* paper as discussed above.

(2) To investigate whether AMPAR and NMDAR changes result from changes in the number of active synapses and/or quantal size, the authors must perform a detailed quantal analysis of electrophysiological recordings and/or provide quantitative measurements of AMPARs and NMDARs at the synaptic level (e.g., through immunostainings or biochemical analysis of synaptosomal fractions).

As described in our response to the introductory comment 2 above, we disagree with this comment, since performing the suggested experiments would be a multi-year project and would not add anything conceptually new to the existing data.

(3) To validate the specificity of their approach, the authors must investigate through immunoblotting and/or RT-qPCR whether the deletion of Cbln1 and/or Cbln2 affect the expression of other Cblns isoforms (Cbln2 and/or Cbln4) as well as Nrxns1/3 and GluD1/2 to ensure that there is no compensatory effect arising from the genetic deletion of Cbln1/2.

We agree and have performed this experiment as described in our response to introductory comment 3 above.

(4) The authors should discuss further their biological model and propose a molecular mechanism by which the Nrxn-Cbln-GluD pathway could directly up-regulate AMPARs or down-regulate NMDARs (depending on the Nrxn and Cblns engaged at each of the connection), without altering synapse formation.

We have added further discussion on these points as requested.

(5) L94-95 – 'Parallel-fiber synapses develop initially normally, but are subsequently lost (Kashiwabuchi et al., 1995; Kurihara et al., 1997; Hirai et al., 2005; Takeuchi et al., 2005)'. This statement is incorrect. The authors are right when saying that the initial parallel fiber synapses develop initially normally, i.e., during the first 10 postnatal days (Kurihara et al., 1997). However, this phase generates only a small fraction of the total number of PF->PC synapses. Actually, most PF-PCs synapses assemble in the following 2 weeks (e.g., Altman et al., J Comp Neurol 1972; Takacs and Hamori, J Neurosci Research, 1994), i.e, when the GluD2 KO mice start displaying abnormal PF synapse number (Kurihara et al., 1997). The statement that PF synapses develop initially normally but are subsequently lost is therefore incorrect.

We have modified this statement to say that at least some PF synapses develop initially normally and are subsequently eliminated. Obviously the existing data do not allow conclusions about whether PF synapses emerging during the first 10 postnatal days develop normally and are eliminated, or simply do not develop – there is no information on this point – the only available information is that at least some PF synapses initially develop.

(6) L114 – 'an RNAi-induced suppression of Cbln2 expression was found to suppress formation of excitatory synapse numbers in the CA1 region of the hippocampus'. This citation is not correct as Tao et al., performed RNAi-mediated GluD1 knock-down +/- Cbln2 overexpression.

The reviewer is correct, and we have amended the statement. Please note that the CA1 region neurons manipulated by Tao et al., do not express endogenous *Cbln1* or *Cbln2*, nor do the presynaptic CA3 inputs onto these neurons express these genes.

(7) L114-L118 – The authors compare studies that used different readouts to monitor synapse number. While Tao et al., quantified spine number and AMPAR / NMDAR-EPSCs in 6d organotypic slices or acute slices from 1 month-old mice, Seigneur and Südhof, as in the present study, quantified average vGluT1 staining intensity from histological sections. The comparison is thus limited and the different methods to quantify synapses should be mentioned.

Again this is correct, except that the Seigneur and Südhof paper also measured protein levels, and that additional Seigneur et al., papers measured the synapse density immunocytochemically.

(8) L121-L122 – 'which is also puzzling since Cbln4 does not bind to GluD1 (Zhong et al., 2017; Cheng et al., 2016)'. Both studies tested the binding of Cblns to GluD2, not GluD1. Moreover, Cheng et al., only looked at Cbln1, not Cbln4. Therefore, the citations are not accurate.

We have corrected the statement and the citations. However, given the fact that GluD1 and GluD2 are highly homologous, that they equally bind to *Cbln1* and *Cbln2*, that they can substitute each other functionally, and that GluD2 doesn’t bind *Cbln4* , it seems highly likely to us that GluD1 will also not bind to Cbln4.

(9) L272 – 'Cbln1 is also expressed in the subiculum, albeit at lower levels'. The images from figure S3 show no Cbln1 mRNA signal (in agreement with Otsuka et al., 2016). The authors should change the image and/or provide some quantitative analysis to support this conclusion.

We have followed this suggestion. Please see our response to introductory point 4 above.

(10) Figure 6C – It is not clear how AMPAR-EPSCs and NMDAR-EPSCs were normalized across recordings for experiments performed in the mPFC since no input-output curve was performed.

We have now added more details of this measurement in the figure legend.

(11) L492 – the argument that 'overactivation of AMPARs may lead to synaptotoxicity thereby explaining synapse loss' is interesting but not supported by any data in the paper nor by relevant citations. It should therefore be discussed further. Actually, the decrease in NMDARs is not in favor of the synaptotoxicity hypothesis.

Agreed. It was offered only as a possible explanation, not as a conclusion. However, NMDARs are not essential for synaptotoxicity since AMPAR-induced depolarization induces calcium-influx via voltage-gated channels.

(12) The discrepancy between the current study and previous reports needs to be further discussed. In particular, Tao et al., reported that Cbln2 overexpression enhances AMPAR and NMDAR EPSCs amplitudes while the present study shows that Cbln2 KO enhances AMPAR EPSCs while decreasing NMDAR-current. The surprising similar effect of Cbln2 overexpression and KO on AMPARs-EPSCs needs to be addressed.

Agreed. We were also surprised by our totally different observations, but please note that the CA1 neurons studied by Tao et al., do not detectably express either *Cbln1* or *Cbln2*, nor do the CA3 input neurons. It is possible that overexpression of *Cbln2* in neurons that don’t normally express it triggers unpredictable signaling pathways.

[Editors’ note: further revisions were suggested prior to acceptance, as described below.]

Reviewer #3 (Recommendations for the authors):In the revised manuscript, the authors have made several improvements and addressed many of my criticisms. However, I still have some concerns about the statements made in the introduction regarding the cerebellum.L96-99: "parallel fiber synapses develop initially, but subsequently decline, with a 40-50% decrease in parallel-fiber synapses in adult Cbln1 or GluD2 KO mice". This sentence is misleading as it suggests that PF synapses form normally and that the decreased number entirely results from abnormal elimination. Actually, as acknowledged by the authors, the current data do not allow to exclude the possibility that the GluD2 KO impairs the critical phase of PF synapse formation which occurs during the 2nd and 3rd weeks. Therefore, this claim should be toned down.

Agreed. It is correct that we cannot exclude the possibility that the GluD2 KO impairs the critical phase of PF synapse formation which occurs during the 2^nd^ and 3^rd^ weeks. However, they are developing normally at the very beginning as shown in the paper of Kurihara et al., 1997. We have now changed it to “In GluD2 KO mice, parallel-fiber synapses develop initially at least in part, but subsequently decline, with a 40-50% decrease in adult GluD2 KO mice”.

L137: "Even in the cerebellum of Cbln1 KO mice, the observed synapse loss is not accompanied by an equivalent decrease in spine density (Hirai et al., 2005)". In contrast to other circuits like PFC, it is well established that PC spines spontaneously form and can remain 'free' even in the absence of PF inputs. In this regard, PC spines cannot be used as a proxy for synapse number and a direct comparison with PFC is not appropriate.

Thank you. This is an important point. We don’t quite agree with the reviewer’s strong statement “that it is well established that PC spines spontaneously form and can remain ‘free’ even in the absence of PF inputs”. To the best of our knowledge, this has never been shown. To take the reviewer’s opinion into account, however, we have now rephased the incriminated sentence in our manuscript to say

“Even in the cerebellum of *Cbln1* KO mice, the observed synapse loss is not accompanied by an equivalent decrease in spine density (Hirai et al., 2005), and it is unknown whether ‘naked’ spines form by itself or represent the remnants of synapses that have been lost.”.

Overall, while data from the authors do support that Nrx-Cbln-GluD signaling is not required for synapse formation in PFC and subiculum (which does not mean that they do not play any role), I think that the idea that the impact of Cbln / GluD2 deletion on PF synapse number in PCs primarily results from increased elimination rather than defect in synapse formation is only weakly supported by published data and appears overstated through the manuscript.

This is an important point of discussion since the opinions on this point clearly differ among scientists. In our manuscript, we emphasized that cerebellins are essential for regulating synaptic properties but not for synapse number in PFC and subiculum. Indeed, deletions of Cbln1 and GluD2 cause a significant, albeit partial, loss of synapses in cerebellum. This loss may be due to a lack of synapse formation, but the literature shows that at least some of the loss is due to an elimination of synapses that have been formed.

Since the cerebellum is not a focus of our study, we have left this question open.